# On Optimal Interpolation in Linear Regression

**Eduard Oravkin**
Department of Statistics
University of Oxford
eduard.oravkin@stats.ox.ac.uk

**Patrick Rebeschini**
Department of Statistics
University of Oxford
patrick.rebeschini@stats.ox.ac.uk

## Abstract

Understanding when and why interpolating methods generalize well has recently been a topic of interest in statistical learning theory. However, systematically connecting interpolating methods to achievable notions of optimality has only received partial attention. In this paper, we investigate the question of what is the optimal way to interpolate in linear regression using functions that are linear in the response variable (as the case for the Bayes optimal estimator in ridge regression) and depend on the data, the population covariance of the data, the signal-to-noise ratio and the covariance of the prior for the signal, but do not depend on the value of the signal itself nor the noise vector in the training data. We provide a closed-form expression for the interpolator that achieves this notion of optimality and show that it can be derived as the limit of preconditioned gradient descent with a specific initialization. We identify a regime where the minimum-norm interpolator provably generalizes arbitrarily worse than the optimal response-linear achievable interpolator that we introduce, and validate with numerical experiments that the notion of optimality we consider can be achieved by interpolating methods that only use the training data as input in the case of an isotropic prior. Finally, we extend the notion of optimal response-linear interpolation to random features regression under a linear data-generating model that has been previously studied in the literature.

## 1 Introduction

Establishing mathematical understanding for the good generalization properties of interpolating methods, i.e. methods that fit the training data perfectly, has attracted significant interest in recent years. Motivated by the quest to explain the generalization performance of neural networks which have zero training error, for example even on randomly corrupted data (Zhang et al., 2017), this area of research has established results for a variety of models. For instance, in kernel regression, Liang and Rakhlin (2020) provide a data-dependent upper bound on the generalization performance of the minimum-norm interpolator. By analyzing the upper bound, they show that small generalization error of the minimum-norm interpolator occurs in a regime with favourable curvature of the kernel, particular decay of the eigenvalues of the kernel and data population covariance matrices and, importantly, in an overparametrized setting. In random features regression, Mei and Montanari (2019) showed that for large signal-to-noise ratio and in the limit of large overparametrization, the optimal regularization is zero, i.e. the optimal ridge regressor is an interpolator. Liang and Sur (2020) characterized the precise high-dimensional asymptotic generalization of interpolating minimum-$\ell_1$-norm classifiers and boosting algorithms which maximize the $\ell_1$ margin. Bartlett et al. (2020) isolated a setting of benign overfitting in linear regression, dependent on notions of effective rank of the population covariance matrix, in which the minimum-norm interpolator has small generalization error. Similarly, this regime of benign overfitting occurs with large overparametrization.

Linear models, in particular, provide a fundamental playground to understand interpolators. On the one hand, in overparametrized regimes, interpolators in linear models are seen to reproduce

35th Conference on Neural Information Processing Systems (NeurIPS 2021).

stylized phenomena observed in more general models. For example, the double descent phenomenon, which was first empirically observed in neural networks (Belkin et al., 2019), has also featured in linear regression (Hastie et al., 2019). On the other hand, neural networks are known to be well-approximated by linear models in some regimes. For example, with specific initialization and sufficient overparametrization, two-layer neural networks trained with gradient descent methods are well-approximated by a first-order Taylor expansion around their initialization (Chizat et al., 2019). This linear approximation can be split into a random features component and a neural-tangent component. The random features model, a two-layer neural network with randomly initialized first layer which is fixed during training, shares similar generalization behavior with the full neural network (Bartlett et al., 2021), and as such, the random features model provides a natural stepping stone towards tackling a theoretical understanding of neural networks.

A major focus of the interpolation literature has so far been to theoretically study if and when interpolating methods based on classical techniques such as ridge regression and gradient descent can have optimal or near-optimal generalization (Bartlett et al., 2021). However the question of understanding which interpolators are best, and designing data-dependent schemes to implement them, seems to have received only partial attention. Work investigating which interpolators are optimal in linear regression includes Muthukumar et al. (2019), where the authors constructed the best-possible interpolator, i.e. a theoretical device which uses knowledge of the true parameter and training noise vector to establish a fundamental limit on how well *any* interpolator in linear regression can generalize. When the whitened features are sub-Gaussian, this fundamental limit is lower bounded by a term proportional to $n/d$, up to an additive constant and with high probability, which is small only in the regime of large overparametrization. Here, $n$ and $d$ are the size and the dimension of the data. While this interpolator provides the best-possible generalization error, the interpolator is not implementable in general, as one would need access to the realization of the true data-generating parameter $w^\star$ and the realization of the noise in the training data. Rangamani et al. (2020) studied generalization of interpolators in linear regression and showed that the minimum-norm interpolator minimizes an *upper* bound on the generalization error related to stability. In (Mourtada, 2020), it was shown that the minimum-norm interpolator is minimax optimal over any choice of the true parameter $w^\star \in \mathbb{R}^d$, distributions of the noise with mean $0$ and bounded variance, and for a fixed nondegenerate distribution of the features. Amari et al. (2021) computed the asymptotic risk of interpolating preconditioned gradient descent in linear regression and investigated the role of its implicit bias on generalization. In particular, they identified the preconditioning which leads to optimal asymptotic (as $d/n \to \gamma > 1$ with $n, d \to \infty$) bias and variance, *separately*, among interpolators of the form $w = PX^T(XPX^T)^{-1}y$ for some matrix $P$, where $X \in \mathbb{R}^{n \times d}$ is the data matrix, $y \in \mathbb{R}^n$ is the response vector. They showed that, within this class of interpolators, using the inverse of the population covariance matrix of the data as preconditioning achieves optimal asymptotic variance. However, the interpolator with optimal *risk* is not given.

In this paper, we study the question of what is the optimal way to interpolate in overparametrized linear regression by procedures that do not use the realization of the true parameter generating the data, nor the realization of the training noise. The motivation for studying this question is twofold. First, in designing new ways to interpolate that are directly related to notions of optimality in linear models, we hope to provide a stepping stone to designing new ways to interpolate in more complex models, such as neural networks. Second, our results illustrate that there can be arbitrarily large differences in the generalization performance of interpolators, in particular considering the minimum-norm interpolator as a benchmark. This is a phenomenon that does not seem to have received close attention in the literature and may spark new interest in designing interpolators connected to optimality.

We consider the family of interpolators that can be achieved as an arbitrary function $f$ of the data, population covariance, signal-to-noise ratio and the prior covariance such that $f$ is linear in the response variable $y$ (as the case for the Bayes optimal estimator in ridge regression). We call such interpolators *response-linear achievable* (see Definition 3). We also introduce a natural notion of optimality that assumes that the realization of true data-generating parameter and the realization of the noise in the training data are unknown. Within this class of interpolators and under this notion of optimality, we theoretically compute the optimal interpolator and show that this interpolator is achieved as the implicit bias of a preconditioned gradient descent with proper initialization. We refer to this interpolator as the *optimal response-linear achievable interpolator*.

Could it be that the commonly used minimum-norm interpolator is good enough so that the benefit of finding a better interpolator is negligible? We illustrate that the answer to this question is, in general,

no. In particular, we construct an example in linear regression where the minimum-norm interpolator has arbitrarily worse generalization than the optimal response-linear achievable interpolator. Here, the variance (hence also generalization error) of the minimum-norm interpolator diverges to infinity as a function of the eigenvalues of the population covariance matrix, while the generalization error of the optimal response-linear achievable interpolator stays bounded, close to the optimal interpolator, i.e. the theoretical device of Muthukumar et al. (2019) which uses the value of the signal and noise.

The optimal response-linear achievable interpolator uses knowledge of the population covariance matrix of the data (similarly as in Amari et al. (2021)), the signal-to-noise ratio, and the covariance of the true parameter (on which we place a prior distribution). Is it the case that the better performance of our interpolator is simply a consequence of this population knowledge? We provide numerical evidence that shows that the answer to this question is, in general, no. In particular, we construct an algorithm to approximate the optimal response-linear achievable interpolator which does not require any prior knowledge of the population covariance or the signal-to-noise ratio and uses only the training data $X$ and $y$, and we empirically observe that this new interpolator generalizes in a nearly identical way to the optimal response-linear achievable interpolator.

Finally, we show that the concept of optimal response-linear achievable interpolation can be extended to more complex models by providing analogous results for a random features model under the same linear data-generating regime as also considered in (Mei and Montanari, 2019), for instance.

## 2 Problem setup

In this paper we investigate overparametrized linear regression. We assume there exists $w^\star \in \mathbb{R}^d$ (unknown) so that $y_i = \langle w^\star, x_i \rangle + \xi_i$ for $i \in \{1, \dots, n\}$, with i.i.d. noise $\xi_i \in \mathbb{R}$ (unknown) such that $\mathbb{E}(\xi_i) = 0, \mathbb{E}(\xi_i^2) = \sigma^2$ and i.i.d. features $x_i \in \mathbb{R}^d$ that follow a distribution $\mathcal{P}_x$ with mean $\mathbb{E}(x_i) = 0$ and covariance matrix $\mathbb{E}(x_i x_i^T) = \Sigma$. We store the features in a random matrix $X \in \mathbb{R}^{n \times d}$ with rows $x_i \in \mathbb{R}^d$, the response variable in a random vector $y \in \mathbb{R}^n$ with entries $y_i \in \mathbb{R}$, and the noise in a random vector $\xi \in \mathbb{R}^n$ with entries $\xi_i \in \mathbb{R}$. Throughout the paper we assume that $d \geq n$. We consider the whitened data matrix $Z = X\Sigma^{-\frac{1}{2}} \in \mathbb{R}^{n \times d}$, whose rows satisfy $\mathbb{E}(z_i z_i^T) = I_d$, where $I_d \in \mathbb{R}^{d \times d}$ is the identity matrix. We place a prior on the true parameter in the form $w^\star \sim \mathcal{P}_{w^\star}$ such that $\mathbb{E}(w^\star) = 0$ and $\mathbb{E}(w^\star w^{\star T}) = \frac{r^2}{d}\Phi$. Here, $\Phi$ is a positive definite matrix and $r^2$ is the signal. We sometimes abuse terminology and refer to $\Phi$ as the covariance of the prior even though $\frac{r^2}{d}\Phi$ is the covariance matrix. Our results will be proved in general, but for the sake of exposition it can be assumed that $x_i \sim \mathcal{N}(0, \Sigma)$, $\xi \sim \mathcal{N}(0, \sigma^2 I_n)$ and $w^\star \sim \mathcal{N}(0, \frac{r^2}{d}\Phi)$. We also define the signal-to-noise ratio $\delta = r^2/\sigma^2$ and consider the squared error loss $\ell : (x, y) \in \mathbb{R}^2 \mapsto (x - y)^2$. Througout the paper, we assume the following two technical conditions hold.

**Assumption 1.** $\mathcal{P}_x(x_i \in V) = 0$ for any linear subspace $V$ of $\mathbb{R}^d$ with dimension smaller than $d$.

**Assumption 2.** For all Lebesgue measurable sets $A \subseteq \mathbb{R}^d$, $\nu(A) > 0$ implies $\mathcal{P}_{w^\star}(w^\star \in A) > 0$, where $\nu$ is the standard Lebesgue measure on $\mathbb{R}^d$.

Assumption 1 is needed only so that $\mathrm{rank}(X) = n$ with probability 1 (for a proof see A.4). A sufficient condition is that $\mathcal{P}_x$ has a density on $\mathbb{R}^d$. A sufficient condition for Assumption 2 is that $\mathcal{P}_{w^\star}$ has a positive density on $\mathbb{R}^d$. Now, our goal is to minimize the population risk

$$r(w) = \mathbb{E}_{\widetilde{x},\widetilde{\xi}}\big((\langle w, \widetilde{x}\rangle - \widetilde{y})^2\big),$$

or, equivalently, the excess risk $r(w) - r(w^\star)$. Here, $(\widetilde{x}, \widetilde{y}, \widetilde{\xi})$ is a random variable which follows the distribution of $(x_1, y_1, \xi_1), \dots, (x_n, y_n, \xi_n)$ and is independent from them. Throughout the paper we write $\mathbb{E}_z g(z, \tilde{z})$ to denote the conditional expectation $\mathbb{E}(g(z, \tilde{z})|\tilde{z})$, for two random variables $z$ and $\tilde{z}$ and for a function $g$. The population risk satisfies

$$r(w) = (w - w^\star)^T \Sigma (w - w^\star) + \sigma^2 = \|w - w^\star\|_\Sigma^2 + r(w^\star), \qquad (1)$$

where $\|w\|_\Sigma^2 = w^T \Sigma w$. We define the bias and variance of $w \in \mathbb{R}^d$ by the decomposition

$$\mathbb{E}_{\xi,w^\star} r(w) = B(w) + V(w), \qquad (2)$$

where

$$B(w) = \mathbb{E}_{\xi,w^\star} \|\mathbb{E}(w|w^\star, X) - w^\star\|_\Sigma^2 \qquad V(w) = \mathbb{E}_{\xi,w^\star} \|w - \mathbb{E}(w|w^\star, X)\|_\Sigma^2. \qquad (3)$$

One of the main paradigms to minimize the (unknown) population risk is based on minimizing the empirical risk $R(w) = \frac{1}{n} \sum_{i=1}^n (\langle w, x_i \rangle - y_i)^2 = \frac{1}{n} \sum_{i=1}^n \ell(w^T x_i, y_i)$ (Vapnik, 1995). In our setting, minimizing the empirical risk is equivalent to finding $w \in \mathbb{R}^d$ such that $Xw = y$.

## 3  Interpolators

An interpolator is any minimizer of the empirical risk. Let $\mathcal{G}$ be the set of interpolators, which in linear regression can be written as

$$\mathcal{G} = \{w \in \mathbb{R}^d : Xw = y\}.$$

As $\text{rank}(X) = n$ with probability 1, we have $\mathcal{G} \neq \emptyset$ with probability 1. In linear regression, the implicit bias of gradient descent initialized at 0 is the minimum-norm interpolator (Gunasekar et al., 2018). We define the minimum-norm interpolator by

$$w_{\ell_2} = \operatorname*{arg\,min}_{w \in \mathbb{R}^d : Xw = y} \|w\|_2^2 = X^\dagger y,$$

where $X^\dagger \in \mathbb{R}^{n \times d}$ is the Moore-Penrose pseudoinverse (Penrose, 1955). As $\text{rank}(X) = n$, we can also write $X^\dagger = X^T(XX^T)^{-1}$. The second interpolator of interest is a purely theoretical device, previously used in (Muthukumar et al., 2019) to specify a fundamental limit to how well any interpolator in linear regression can generalize.

**Definition 1.** The *best possible interpolator* is defined as

$$W_b = \operatorname*{arg\,min}_{w \in \mathcal{G}} r(w).$$

We can write

$$W_b = \operatorname*{arg\,min}_{w \in \mathbb{R}^d : Xw = y} \|\Sigma^{\frac{1}{2}}(w - w^\star)\|_2^2,$$

and after a linear transformation and an application of a result on approximate solutions to linear equations (Penrose, 1956), we obtain

$$W_b = w^\star + \Sigma^{-\frac{1}{2}}(X\Sigma^{-\frac{1}{2}})^\dagger \xi. \qquad (4)$$

We notice that the best possible interpolator fits the signal perfectly by having access to the true parameter $w^\star$ and fits the noise through the term $\Sigma^{-\frac{1}{2}}(X\Sigma^{-\frac{1}{2}})^\dagger \xi$ by having access to the noise vector $\xi$ in the training data. In general, this interpolator cannot be implemented as it requires access to the unknown quantities $w^\star$ and $\xi$. We are interested in interpolators which can be achieved by some algorithm using the data $X$ and $y$.

**Definition 2.** We define an estimator $w \in \mathbb{R}^d$ to be *achievable* if there exists a function $f$ such that $w = f(X, y, \Sigma, \Phi, \delta)$.

In our definition of achievability, we allow for knowledge of the population data covariance, the signal-to-noise ratio, and the prior covariance to define a fundamental limit to what generalization performance can be achieved also without access to these quantities, and we later empirically show that we can successfully approach this limit using only the knowledge of the training data $X$ and $y$, in considered examples (see Section 5.1). Moreover, our theory is also useful in situations when one has access to some prior information about the regression problem which they can incorporate into an estimate of $\Sigma$, $\delta$, $\Phi$ (for example, one may know the components of $x_i$ are independent and hence $\Sigma$ is diagonal) and hence it is relevant to consider a broader class than $w = f(X, y)$.

**Definition 3.** We define the set of *response-linear achievable estimators* by

$$\mathcal{L} = \{\, w \in \mathbb{R}^d : \exists f \text{ such that } w = f(X, y, \Sigma, \Phi, \delta) \text{ where } y \in \mathbb{R}^n \mapsto f(X, y, \Sigma, \Phi, \delta) \text{ is linear} \,\}.$$

Linearity of $y \in \mathbb{R}^n \mapsto f(X, y, \Sigma, \Phi, \delta)$ is equivalent to $f(X, y, \Sigma, \Phi, \delta) = g(X, \Sigma, \Phi, \delta)y$, where $g$ is any function which has image in $\mathbb{R}^{d \times n}$. The notion of optimality that we introduce is that of the optimal response-linear achievable interpolator, which is the interpolator that minimizes the expected risk in the class $\mathcal{L}$.

**Definition 4.** We define the *optimal response-linear achievable interpolator* by

$$w_O = \underset{w \in \mathcal{G} \cap \mathcal{L}}{\arg\min} \, \mathbb{E}_{\xi, w^\star} r(w) - r(w^\star). \tag{5}$$

## 4 Main results

By definition, the interpolator $w_O$ has the smallest expected risk among all response-linear achievable interpolators. Our first contribution is the calculation of its exact form.

**Proposition 1.** The optimal response-linear achievable interpolator satisfies

$$w_O = \left( \frac{\delta}{d} \Phi X^T + \Sigma^{-\frac{1}{2}} (X \Sigma^{-\frac{1}{2}})^\dagger \right) \left( I_n + \frac{\delta}{d} X \Phi X^T \right)^{-1} y. \tag{6}$$

For an isotropic prior $\Phi = I_d$, $w_O$ depends only on the population covariance $\Sigma$ and the signal-to-noise ratio $\delta$ so that $w_O$ can be approximated using estimators of these quantities, which is what we do in Sections 5 and A.8. Even if $\Phi \neq I_d$, one might have some information about the prior covariance, which can be incorporated into an estimate $\widehat{\Phi}$ and used instead of $\Phi$. However, even if no such estimate is available, in Section A.9 we empirically show that, in our examples, using $\widehat{\Phi} = I_d$ when $\Phi \neq I_d$ has a small effect on generalization.

Secondly, using results of Gunasekar et al. (2018) on the implicit bias of converging mirror descent, we show that the optimal response-linear interpolator is the limit of gradient descent preconditioned by the inverse of the population covariance, provided that it converges and is suitably initialized.

**Proposition 2.** The optimal response-linear achievable interpolator is the limit of preconditioned gradient descent

$$w_{t+1} = w_t - \eta_t \Sigma^{-1} \nabla R(w_t), \tag{7}$$

provided that the algorithm converges, initialized at

$$w_0 = \frac{\delta}{d} \Phi X^T \left( I_n + \frac{\delta}{d} X \Phi X^T \right)^{-1} y. \tag{8}$$

The interpolator $w_O$ does not have the smallest bias or the smallest variance in the bias-variance decomposition $\mathbb{E}_{\xi, w^\star} \, r(w) = B(w) + V(w)$, but rather achieves a balance. This is related to the results of (Amari et al., 2021). Their setting looks at interpolators achieved as the limit of preconditioned gradient descent in linear regression (preconditioned with some matrix $P$) and initialized at 0. Such interpolators can be written as $w = PX^T(XPX^T)^{-1}y$. For these interpolators, they compute the risk of $w$, separate the risk into a variance and a bias term and using random matrix theory they find what the variance and bias terms converge to when $d \to \infty, n \to \infty$ in a way such that $d/n \to \gamma > 1$. For these calculations to hold, they assume that the spectral distribution of $(\Sigma_d)_{d \in \mathbb{N}}$ converges weakly to a distribution supported on $[c, C]$ for some $c, C > 0$. Then, after obtaining the limiting variance and bias, they prove which matrices $P$ minimize these limits *separately* (not their sum, which is the overall asymptotic risk).

We approach the problem from the other direction. That is, we do not a priori consider interpolators that can be achieved as limits of specific algorithms, but we directly look at which interpolator minimizes the risk as a whole (not bias and variance separately). Only after computing the optimal response-linear interpolator, we show in Proposition 2 that the interpolator *is* in fact the limit of preconditioned gradient descent, however with a specific initialization. Our results hold for every finite $d \geq n$ and we do not put assumptions on the eigenvalues or the spectral distribution of $\Sigma$.

In particular, we can recover the results of (Amari et al., 2021) as a special case of Proposition 1. If we take the signal-to-noise ratio $\delta \to 0$ (by taking $r^2 \to 0$) in Proposition 1, we obtain the matrix $P$

which achieves optimal variance and if we take $\delta \to \infty$ (by taking $\sigma^2 \to 0$), we obtain the matrix $P$ which achieves optimal bias. Moreover, we provide a further extension in Proposition 3.

We show that the preconditioned gradient descent $w_{t+1} = w_t - \eta_t \Sigma^{-1} \nabla R(w_t)$ achieves optimal variance among *all* interpolators when initialized at *any* deterministic $w_0$ and for *any* finite $d, n \in \mathbb{N}$.

**Proposition 3.** The limit of preconditioned gradient descent $w_{t+1} = w_t - \eta_t \Sigma^{-1} \nabla R(w_t)$ initialized at a deterministic $w_0 \in \mathbb{R}^d$, provided that it converges, satisfies

$$\lim_{t \to \infty} w_t = \arg\min_{w \in \mathcal{G}} V(w). \tag{9}$$

We note that the optimal variance is achieved among all interpolators, not only among response-linear achievable interpolators.

A natural question to ask is whether the optimal response-linear achievable interpolator $w_O$ provides a significant benefit compared to other interpolators. A second question is whether we can successfully approximate the optimal response-linear achievable interpolator without knowledge of the population covariance $\Sigma$ and the signal-to-noise ratio $\delta$. In the following section, we illustrate that both the interpolator with optimal variance and the interpolator with optimal bias can generalize arbitrarily badly in comparison to $w_O$ as a function of the eigenvalues of the population covariance. In the same regimes where this happens, we present numerical evidence that we can successfully approximate $w_O$ by an empirical interpolator $w_{Oe}$ without any prior knowledge of $\Sigma$ or $\delta$ by using the Graphical Lasso estimator (Friedman et al., 2007) of the covariance matrix $\Sigma$ and choosing the empirical estimate of $\delta$ by crossvalidation on a subset of the data.

## 5 Comparison of interpolators

First, we present an example where the minimum-norm interpolator $w_{\ell_2}$ generalizes arbitrarily worse than the best response-linear achievable interpolator $w_O$. Second, we give an example where an interpolator with optimal variance generalizes arbitrarily worse than $w_O$. This shows that arbitrarily large differences in test error are possible within the class of estimators with zero training error.

In the examples, we consider a setting where $x_i \sim \mathcal{N}(0, \Sigma)$ and $w^\star \sim \mathcal{N}(0, \frac{r^2}{d}\Phi)$. Therefore, throughout Section 5 we assume $\mathcal{P}_x = \mathcal{N}(0, \Sigma)$ and $\mathcal{P}_{w^\star} = \mathcal{N}(0, \frac{r^2}{d}\Phi)$. Before presenting these examples, we discuss approximating $w_O$ by an interpolator, $w_{Oe}$, which uses only the data $X$ and $y$.

### 5.1 Empirical approximation

The interpolator $w_O$ is the limit of the algorithm

$$w_{t+1} = w_t - \eta_t \Sigma^{-1} \nabla R(w_t). \tag{10}$$

The population covariance $\Sigma$ is required to run this algorithm. However, the matrix $\Sigma$ is usually unknown in practice. One may want to estimate $\Sigma$. However, if one replaces $\Sigma$ by $\widetilde{\Sigma} = X^T X/n + \lambda I_d$ (with $\lambda \geq 0$), then the limit of (10) is the same as the limit of gradient descent (provided that both algorithms converge). This is because, using the singular value decomposition of $X$, one can show

$$\widetilde{\Sigma}^{-\frac{1}{2}}(X\widetilde{\Sigma}^{-\frac{1}{2}})^\dagger y = X^\dagger y.$$

The preconditioned gradient update $w_{t+1} - w_t = \eta_t P^{-1} \nabla R(w_t)$ has to *not* belong to $\text{Im}(X^T)$ in order to *not* converge to the minimum-norm interpolator. Hence, using $P = \widetilde{\Sigma} = X^T X/n + \lambda I_d$ (for example, also the Ledoit-Wolf shrinkage approximation (Ledoit and Wolf, 2004)) in preconditioned gradient descent removes the benefit of preconditioning in terms of generalization of the limit.

We use the Graphical Lasso approximation (Friedman et al., 2007). We empirically observe that in the examples considered in this paper (Figures 1, 2, 3, 4, 5, 6) using the Graphical Lasso covariance $\Sigma_e$ instead of $\Sigma$ has nearly no effect on generalization. Under specific assumptions, Ravikumar et al. (2011) provide some convergence guarantees of the Graphical Lasso.

In regards to approximating the signal-to-noise ratio $\delta$, we choose $\delta_e$ that minimizes the crossvalidated error on random subsets of the data. In this way, we arrive at the interpolator

$$w_{Oe} = \left( \frac{\delta_e}{d} X^T + \Sigma_e^{-\frac{1}{2}}(X\Sigma_e^{-\frac{1}{2}})^\dagger \right) \left( I_n + \frac{\delta_e}{d} X X^T \right)^{-1} y, \tag{11}$$

which approximates $w_O$ and is a function of only $X$ and $y$. We note that the interpolator $w_{Oe}$ uses $I_d$ in place of the prior covariance matrix.

In the experiments (Figures 1, 2, 3, 4, 5, 6) we used the Graphical Lasso implementation of scikit-learn (Pedregosa et al., 2011) with parameter $\alpha = 0.25$ ($\alpha$ can also be crossvalidated for even better performance) and in estimating $\delta$, for each $\delta_e$ in $\{0.1, 0.2, \ldots, 1, 2, \ldots, 10\}$, we computed the validation error on a random, unseen tenth of the data and averaged over 10 times. The $\delta_e$ with smallest crossvalidated error was chosen.

## 5.2 Random matrix theory concepts

For presenting the discussed examples we need to review some concepts from random matrix theory.

**Definition 5.** For a symmetric matrix $\Sigma \in \mathbb{R}^{d \times d}$ with eigenvalues $\lambda_1 \geq \lambda_2 \geq \cdots \geq \lambda_d \geq 0$ we define its spectral distribution by $\mathcal{F}_\Sigma(x) = \frac{1}{d} \sum_{i=1}^d \mathbb{1}_{[\lambda_i, \infty)}(x)$.

The following assumptions will be occasionally considered for the covariance matrix $\Sigma$.

**Assumption 3.** There exists $k_{\max} > 0$ such that $\lambda_{\max}(\Sigma) \leq k_{\max}$ uniformly for $d \in \mathbb{N}$.

**Assumption 4.** There exists $k_{\min} > 0$ such that $k_{\min} \leq \lambda_{\min}(\Sigma)$ uniformly for $d \in \mathbb{N}$.

**Assumption 5.** The spectral distribution $\mathcal{F}_\Sigma$ of the covariance matrix $\Sigma$ converges weakly to a distribution $\mathcal{H}$ supported on $[0, \infty)$.

Marčenko and Pastur (1967) showed that there exists a distribution $\widetilde{\mathcal{F}}_\gamma$ such that $\mathcal{F}_{\frac{Z^T Z}{n}} \longrightarrow \widetilde{\mathcal{F}}_\gamma$, weakly, with probability 1 as $n \to \infty, d \to \infty$ with $d/n \to \gamma$. In our discussion, $x_i = \Sigma^{\frac{1}{2}} z_i$, where $z_i \sim \mathcal{N}(0, I_d)$ independently. Then, under Assumption 5, it can be shown that the spectral distribution of $\widehat{\Sigma} = X^T X/n = \Sigma^{\frac{1}{2}} Z^T Z \Sigma^{\frac{1}{2}}/n$ converges weakly, with probability 1 to a distribution supported on $[0, \infty)$, which we denote by $F_\gamma$, see e.g. (Silverstein and Choi, 1995). Similar arguments also show that the spectral distribution of $XX^T/n \in \mathbb{R}^{n \times n}$ converges weakly, with probability 1.

**Definition 6.** For a distribution $\mathcal{F}$ supported on $[0, \infty)$, we define the *Stieltjes transform of $\mathcal{F}$*, for any $z \in \mathbb{C} \setminus \mathbb{R}^+$ by

$$m_\mathcal{F}(z) = \int_0^\infty \frac{1}{\lambda - z} d\mathcal{F}(\lambda).$$

The weak convergence of the spectral distribution of $\widehat{\Sigma}$ to $\mathcal{F}_\gamma$ is equivalent to $m_{\widehat{\Sigma}}(z) \to m(z)$ and $m_{XX^T/n}(z) \to v(z)$ almost surely for all $z \in \mathbb{C} \setminus \mathbb{R}^+$, where $m$ and $v$ are the Stieltjes transforms of $\mathcal{F}_\gamma$ and the limiting spectral distribution of $XX^T/n$, respectively (see e.g. Proposition 2.2 of (Hachem et al., 2007)). We call $v$ the companion Stieltjes transform of $\mathcal{F}_\gamma$.

## 5.3 Diverging variance of interpolator with optimal bias

Using $w_0 = 0$ in Proposition 3, we choose the interpolator with optimal variance to be (see A.3)

$$w_V = \Sigma^{-\frac{1}{2}} (X \Sigma^{-\frac{1}{2}})^\dagger y. \tag{12}$$

When $\Phi = I_d$, the interpolator with best bias among response-linear achievable interpolators is the minimum-norm interpolator (see Section A.5). We identify an example where the minimum-norm interpolator $w_{\ell_2}$ generalizes arbitrarily worse than the best response-linear achievable interpolator $w_O$. For this, we exploit results of Hastie et al. (2019) on computing the asymptotic risk of the minimum-norm interpolator. They show that if $\Phi = I_d$, under Assumptions 3, 4, 5 and if $\frac{d}{n} \to \gamma > 1$ with $n \to \infty, d \to \infty$ then with probability 1,

$$\mathbb{E}_{\xi, w^\star} r(w_{\ell_2}) - r(w^\star) \longrightarrow \frac{r^2}{\gamma v(0)} + \sigma^2 \left( \frac{v'(0)}{v(0)^2} - 1 \right), \tag{13}$$

where $v$ is the companion Stieltjes transform introduced in Section 5.2. In comparison, similarly as in (Amari et al., 2021), the asymptotic risk of the best variance estimator $w_V$ satisfies that under

Assumption 3 and 5, if $n, d \to \infty$ with $\frac{d}{n} \to \gamma > 1$ then with probability 1 we have that

$$\lim_{d \to \infty} \mathbb{E}_{\xi, w^*} r(w_V) - r(w^\star) = r^2 \frac{\gamma - 1}{\gamma} \int_0^\infty s \, d\mathcal{H}(s) + \frac{\sigma^2}{\gamma - 1}. \tag{14}$$

An alternative way is to write $\int_0^\infty s \, d\mathcal{H}(s) = \lim_{d \to \infty} \text{Tr}(\Sigma)$. This result follows by an application of Theorem 1 of (Rubio and Mestre, 2011), which is in the supplementary material for completeness.

Now we find a regime of covariance matrices $\Sigma$, for which the variance term of the minimum-norm solution, $V_{\ell_2} = \sigma^2(\frac{v'(0)}{v(0)^2} - 1)$, diverges to infinity, while the risk of $w_O$ stays bounded and close to optimal. For this, we consider a generalization of the spike model of covariance matrices (Baik and Silverstein, 2006; Johnstone, 2001), which is a fundamental model in statistics. Here $\Sigma = \text{diag}(\rho_1, \ldots, \rho_1, \rho_2, \ldots, \rho_2) \in \mathbb{R}^{d \times d}$, where the number of $\rho_1$s is $d \cdot \psi_1$ with $\psi_1 \in [0, 1]$. This model was also considered in (Richards et al., 2021) where it is called the strong weak features model. In this regime, it is possible to explicitly calculate the companion Stieltjes transform $v(0)$ and $v'(0)$ of (13). In the case that $\gamma = 2, \psi_1 = 1/2$ we have

$$V_{\ell_2} = \sigma^2 \left( \frac{v'(0)}{v(0)^2} - 1 \right) = \frac{\sigma^2}{2} \left( \sqrt{\frac{\rho_1}{\rho_2}} + \sqrt{\frac{\rho_2}{\rho_1}} + 2 \right). \tag{15}$$

If we fix $\rho_1 = 1$ and take $\rho_2 \to 0$, then the variance term $V_{\ell_2}$ diverges to infinity. This also means that the asymptotic risk of the minimum-norm interpolator diverges to infinity. Moreover, the asymptotic risk of $w_V$ in (14) evaluates to

$$\lim_{d \to \infty} \mathbb{E}_{\xi, w^*} r(w_V) - r(w^\star) = \left( \psi_1 \rho_1 + (1 - \psi_1) \rho_2 \right) \left( 1 - \frac{1}{\gamma} \right) + \frac{\sigma^2}{\gamma - 1}. \tag{16}$$

In addition, by construction of $w_O$, we know that $\mathbb{E}_{\xi, w^\star} r(w_O) \leq \mathbb{E}_{\xi, w^\star} r(w_V)$ and therefore the asymptotic limit of $\mathbb{E}_{\xi, w^\star} r(w_O) - r(w^\star)$, as $d/n \to \gamma > 1$, stays bounded by (16) as $\rho_2 \to 0$. The expected generalization error in the setting described above is illustrated in Figure 1.

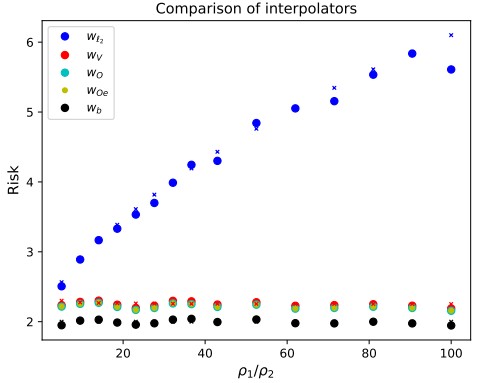
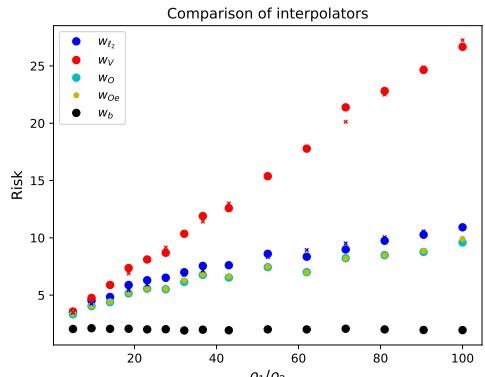

Figure 1: Plot of $\mathbb{E}_\xi r(w)$ (points) for $w \in \{w_{\ell_2}, w_V, w_O, w_{Oe}, w_b\}$ along with predictions (crosses) from (14) and (13) in the strong weak features model with $r^2 = 1, \sigma^2 = 1, \gamma = 2, \psi_1 = 1/2, n = 3000$ and $\rho_1 = 1, \rho_2 \to 0$.

Figure 2: Plot of $\mathbb{E}_\xi r(w)$ (points) for $w \in \{w_{\ell_2}, w_V, w_O, w_{Oe}, w_b\}$ along with predictions (crosses) from (14) and (13) in the strong weak features model with $r^2 = 1, \sigma^2 = 1, \gamma = 2, \psi_1 = 1/2, n = 3000$ and $\rho_2 = 1, \rho_1 \to \infty$.

We note that the empirical estimator $w_{Oe}$ (yellow points), which is a function of only the training data $X$ and $y$ and does not use the population covariance $\Sigma$ or the signal-to-noise ratio $\delta$, performs almost identically to the optimal response-linear achievable interpolator $w_O$ (cyan points).

In this example, we chose $\gamma = 2$ and $\psi_1 = 1/2$ deliberately. One does not achieve diverging variance for an arbitrary choice of $\gamma$ and $\psi_1$. However, for any $\gamma > 1$ such that $\gamma \psi_1 = 1$, the phenomenon of Figure 1 holds (see A.7 of the supplementary material).

## 5.4 Diverging bias of interpolator with optimal variance

Now, we illustrate a regime where the best variance interpolator $w_V$ generalizes arbitrarily worse than $w_O$. In the same strong and weak features covariance model described above in Section 5.3, when $\gamma = 2$ and $\psi_1 = 1/2$, if we instead have $\rho_1 \to \infty$ and $\rho_2 = 1$, then the asymptotic risk (16) diverges to infinity linearly. However, the variance of the minimum-norm interpolator in (15) diverges only like $\sqrt{\rho_1}$. Moreover, the bias term satisfies

$$B_{\ell_2} = \frac{r^2}{\gamma v(0)} = \frac{r^2}{\gamma}\sqrt{\rho_1 \rho_2},$$

which also diverges like $\sqrt{\rho_1}$. Now, because $\mathbb{E}_{\xi,w^\star} r(w_O) \le \mathbb{E}_{\xi,w^\star} r(w_{\ell_2})$, we have that

$$\lim_{d \to \infty} \mathbb{E}_{\xi,w^\star} r(w_O) \le \frac{r^2}{\gamma}\sqrt{\rho_1 \rho_2} + \frac{\sigma^2}{2}\left( \sqrt{\frac{\rho_1}{\rho_2}} + \sqrt{\frac{\rho_2}{\rho_1}} + 3 \right),$$

so that the asymptotic risk of $w_O$ diverges to infinity as $\sqrt{\rho_1}$. We illustrate this in Figure 2.

We notice that the empirical approximation $w_{Oe}$ again performs in a nearly identical way to the optimal response-linear achievable interpolator $w_O$. Moreover, importantly, we note that $w_V$ and $w_O$ are limits of the same algorithm, $w_{t+1} = w_t - \eta_t \Sigma^{-1} \nabla R(w_t)$, only with different initialization. Hence, this shows that different initialization of the same optimization algorithm can have an arbitrarily large influence on generalization through implicit bias.

# 6 Random features regression

The concept of optimal interpolation as a function which is linear in the response variable, is general and can be extended beyond linear models. We present an extension of Proposition 1 to the setting of random features regression. Random features models were introduced as a random approximation to kernel methods (Rahimi and Recht, 2008) and can be viewed as a two-layer neural network with first layer randomly initialized and fixed as far as training is concerned. They can be shown to approximate neural networks in certain regimes of training and initialization and hence are often considered in the literature as a first step to address neural networks (e.g. (Jacot et al., 2018)). We consider data generated in the same way as before, $y_i = \langle x_i, w^\star \rangle + \xi_i$, and the model to be a two-layer neural network $f_a : \mathbb{R}^d \ni x \mapsto a^T \sigma(\Theta x / \sqrt{d})$, where the first layer $\Theta \in \mathbb{R}^{N \times d}$ is randomly initialized. This setting, along with $x_i$ and rows of $\Theta$ belonging to the sphere $\mathbb{S}^{d-1}(\sqrt{d})$ with radius $\sqrt{d}$ in $\mathbb{R}^d$, is often considered in the literature on interpolation of random features models (Mei and Montanari, 2019; Ghorbani et al., 2021). If we analogously define the optimal response-linear achievable interpolator in random features regression by

$$a_O = \arg\min_{a \in \mathcal{G} \cap \mathcal{L}} \mathbb{E}_{\xi, w^\star} r(f_a) - r(w^\star), \tag{17}$$

where here $\mathcal{G} = \{a \in \mathbb{R}^N : Za = y\}$ is the set of interpolators, $Z = \sigma(X\Theta^T/\sqrt{d})$ and $\mathcal{L}$ is the same as in Definition 3, then the following analogue of Proposition 1 holds.

**Proposition 4.** The optimal response-linear achievable interpolator (17) in random features regression satisfies

$$a_O = \Sigma_z^{-1}\left( \Sigma_{zx}\Phi X^T + Z^T \left( Z\Sigma_z^{-1}Z^T \right)^{-1}\left( \frac{d}{\delta}I_n + X\Phi X^T - Z\Sigma_z^{-1}\Sigma_{zx}\Phi X^T \right) \right)\left( \frac{d}{\delta}I_n + X\Phi X^T \right)^{-1} y,$$

Here $\Sigma_z = \mathbb{E}_{\tilde{x}}(\sigma(\Theta\tilde{x}/\sqrt{d})\sigma(\Theta\tilde{x}/\sqrt{d})^T)$ and $\Sigma_{zx} = \mathbb{E}_{\tilde{x}}(\sigma(\Theta\tilde{x}/\sqrt{d})\tilde{x}^T)$ are covariance and cross-covariance matrices, respectively. This interpolator can be again obtained as the implicit bias of preconditioned gradient descent using results of Gunasekar et al. (2018).

**Proposition 5.** The optimal response-linear achievable interpolator (17) in random features regression is the limit of preconditioned gradient descent on the last layer,

$$w_{t+1} = w_t - \eta_t \Sigma_z^{-1} \nabla R(w_t),$$

provided that the algorithm converges, initialized at

$$a_0 = \Sigma_z^{-1}\Sigma_{zx}\Phi X^T \left( \frac{d}{\delta}I_n + X\Phi X^T \right)^{-1} y.$$

In Section A.12, we illustrate the test error of $f_a$, with $a = a_O$ in comparison to the test error for the minimum-norm interpolator $a = a_{\ell_2} = Z^\dagger y$ on a standard example.

# 7 Conclusion

In this paper, we investigated how to design interpolators in linear regression which have optimal generalization performance. We designed an interpolator which has optimal risk among interpolators that are a function of the training data, population covariance, signal-to-noise ratio and prior covariance, but does not depend on the true parameter or the noise, where this function is linear in the response variable. We showed that this interpolator is the implicit bias of a covariance-based preconditioned gradient descent algorithm. We identified regimes where other interpolators of interest are arbitrarily worse using computations of their asymptotic risk as $\frac{d}{n} \to \gamma > 1$ with $d, n \to \infty$.

In particular, we found a regime where the variance term of the minimum-norm interpolator is arbitrarily large compared to our interpolator. This confirms the phenomenon that implicit bias has an important influence on generalization through the choice of optimization algorithm.

We identified a second regime where the interpolator that has best variance is arbitrarily worse than our interpolator. In this second example, both interpolators are the implicit bias of the same algorithm, but with different initialization. This contributes to illustrating that initialization has an important influence on generalization.

We also considered an empirical approximation of the optimal response-linear achievable interpolator, which uses only the training data $X$ and $y$ and does not assume knowledge of the population covariance matrix, the signal-to-noise ratio or the prior covariance and empirically observe that it generalizes in a nearly identical way to the optimal response-linear achievable interpolator in the examples that we consider.

A limitation of this work includes a precise guarantee on the approximation error of the Graphical Lasso for a general covariance matrix $\Sigma$. Some guarantees are in (Ravikumar et al., 2011), however establishing guarantees for a general covariance matrix would be a contribution on its own.

A natural question for future research, which also motivated our work, is how to systematically design new ways of interpolation, which are adapted to the distribution of the data and related to notions of optimality, for more complex overparametrized machine learning models such as neural networks.

# 8 Acknowledgements

The authors would like to thank Dominic Richards, Edgar Dobriban and the anonymous reviewers for valuable insights which contributed to the technical quality of the paper. Eduard Oravkin was part-time employed at the Department of Statistics during a part of this project. Patrick Rebeschini was supported in part by the Alan Turing Institute under the EPSRC grant EP/N510129/1.

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
