# A  Supplementary Material

## A.1  Proof of Propostion 1

We prove that the optimal response-linear achievable interpolator in linear regression is

$$w_O = \left( \frac{\delta}{d} \Phi X^T + \Sigma^{-\frac{1}{2}} (X\Sigma^{-\frac{1}{2}})^\dagger \right) \left( I_n + \frac{\delta}{d} X \Phi X^T \right)^{-1} y.$$

*Proof.* First, we note that as $w_O \in \mathcal{L}$, there exists $\mathbb{R}^{d \times n} \ni Q = Q(X, \Sigma, \Phi, \delta)$ such that $w_O = Qy = QXw^\star + Q\xi$. Therefore, the definition of $w_O$,

$$w_O = \arg\min_{w \in \mathcal{G} \cap \mathcal{L}} \mathbb{E}_{\xi, w^\star} r(w) - r(w^\star),$$

can be restated as

$$w_O = \arg\min_{\substack{w_O = Qy \\ XQy = y}} B(w_O) + V(w_O), \tag{18}$$

where

$$B(w_O) = \frac{r^2}{d} \mathrm{Tr}\big( \Sigma (QX - I_d) \Phi (QX - I_d)^T \big),$$

$$V(w_O) = \sigma^2 \mathrm{Tr}\big( \Sigma Q Q^T \big).$$

**Claim 1.** (18) implies

$$w_O = \arg\min_{\substack{w_O = Qy \\ XQX = X}} B(w_O) + V(w_O). \tag{19}$$

We prove this claim. $XQy = y$ almost surely for all realizations of the data (that is, a.s. for all realizations of $X, \xi, w^\star$) implies

$$0 = \mathbb{E}\big( XQ(Xw^\star + \xi) | X, w^\star \big) - \mathbb{E}\big( Xw^\star + \xi | X, w^\star \big) = X(QX - I_d)w^\star$$

almost surely for all realizations of $X, w^\star$. Therefore,

$$0 = \mathbb{E}_{w^\star}\big( \|X(QX - I_d)w^\star\|_2^2 \mid w^\star \big) = w^{\star T} \mathbb{E}\big( (QX - I_d)^T X^T X (QX - I_d) \big) w^\star \tag{20}$$

almost surely for all realizations of $w^\star$. It follows that $\mathbb{E}\big( (QX - I_d)^T X^T X (QX - I_d) \big) = 0$. This is because, if not, then there exists $v \in \mathbb{R}^d$ and $\epsilon > 0$ such that

$$\forall u \in B_\epsilon(v) \qquad u^T \mathbb{E}\big( (QX - I_d)^T X^T X (QX - I_d) \big) u > 0. \tag{21}$$

Recall that $w^\star \sim \mathcal{P}_{w^\star}$, where $\mathcal{P}_{w^\star}$ is by assumption such that $\nu(A) > 0$ implies $\mathcal{P}_{w^\star}(A) > 0$ for all Lebesgue measurable $A \in \mathbb{R}^d$ (where $\nu$ is the Lebesgue measure). However, as $B_\epsilon(v)$ has positive Lebesgue measure, (21) is hence a contradiction to (20). Finally, as $\|A\| = \sqrt{\mathrm{Tr}(AA^T)}$ is a norm, $\mathbb{E}\big( (QX - I_d)^T X^T X (QX - I_d) \big) = 0$ implies that $X(QX - I_d) = 0$ almost surely. This finishes the proof of the claim.

Now we use Theorem 2 of Penrose (1955) which states that for any matrices $A, B, C$ and $D$, all solutions $B$ to the equation $ABC = D$ can be written as $B = A^\dagger D C^\dagger + S - A^\dagger A S C C^\dagger$ where $S$ is arbitrary. Therefore, $XQX = X$ is equivalent to

$$Q = X^\dagger X X^\dagger + S - X^\dagger X S X X^\dagger = X^\dagger + S - X^\dagger X S$$

for some arbitrary $S \in \mathbb{R}^{d \times n}$. Hence, if we write $Q = Q(S)$ and $w_O = w_O(S)$, (19) is equivalent to an unconstrained optimization problem over $\mathbb{R}^{d \times n}$ in the form

$$w_O = \arg\min_S f(S), \tag{22}$$

where $f(S) = B(w_O(S)) + V(w_O(S))$. Now we show that $f : \mathbb{R}^{d \times n} \to \mathbb{R}$ is strictly convex. Note that the map

$$\mathbb{R}^{d \times n} \ni S \mapsto \Sigma^{\frac{1}{2}} (Q(S)X - I_d) \Phi^{\frac{1}{2}}$$

is affine and nonzero and the map $\mathbb{R}^{d \times d} \ni A \mapsto \mathrm{Tr}(AA^T)$ is strictly convex because $\|A\| = \sqrt{\mathrm{Tr}(AA^T)}$ is a norm. The composition of these two maps is $\mathbb{R}^{d \times n} \ni S \mapsto \frac{d}{r^2}B(w_O(S))$, which is therefore strictly convex. A similar argument proves that $\mathbb{R}^{d \times n} \ni S \mapsto V(w_O(S))$ is strictly convex and hence also $f$ is. Moreover, $f$ is differentiable. Therefore, to find a unique global minimum of $f$, it is enough to find $S^\star \in \mathbb{R}^{d \times n}$ such that $\partial f(S^\star) = 0$. Using tools of matrix calculus we find

$$\partial f(S) = 2(I_d - X^\dagger X)\Sigma A,$$

where

$$A = \left( \sigma^2 S + \frac{r^2}{d}(SX - I_d)\Phi X^T + X^\dagger(I_n - XS)(\sigma^2 I_n + \frac{r^2}{d}X\Phi X^T) \right).$$

Because $\mathbb{R}^d \ni v \mapsto (I_d - X^\dagger X)v$ is the projection onto $\mathrm{Ker}(X) = \mathrm{Im}(X^T)^\perp$, this hints towards finding $S^\star$ such that $A = \Sigma^{-1}X^T B$ for some matrix $B$. This is achieved, for example, if

$$\sigma^2 S^\star + \frac{r^2}{d}(S^\star X - I_d)\Phi X^T = \Sigma^{-1}X^T B$$

and

$$I_n - XS^\star = 0,$$

for some matrix $B$. Putting the two equations together implies $B = \sigma^2(X\Sigma^{-1}X^T)^{-1}$ and hence, using that $\Sigma^{-\frac{1}{2}}(X\Sigma^{-\frac{1}{2}})^\dagger = \Sigma^{-1}X^T(X\Sigma^{-1}X^T)^{-1}$ and $\delta = \frac{r^2}{\sigma^2}$, we have

$$S^\star = \left( \frac{\delta}{d}\Phi X^T + \Sigma^{-\frac{1}{2}}(X\Sigma^{-\frac{1}{2}})^\dagger \right)\left( I_n + \frac{\delta}{d}X\Phi X^T \right)^{-1}.$$

Finaly, because $XS^\star = I_n$, it follows that $Q^\star = X^\dagger + S^\star - X^\dagger XS^\star = S^\star$ and hence

$$w_O = Q^\star y = \left( \frac{\delta}{d}\Phi X^T + \Sigma^{-\frac{1}{2}}(X\Sigma^{-\frac{1}{2}})^\dagger \right)\left( I_n + \frac{\delta}{d}X\Phi X^T \right)^{-1}y.$$

$\square$

## A.2 Proof of Proposition 2

We prove that the optimal response-linear achievable interpolator $w_O$ is the limit of preconditioned gradient descent

$$w_{t+1} = w_t - \eta_t \Sigma^{-1}\nabla R(w_t), \tag{23}$$

provided that the algorithm converges, initialized at

$$w_0 = \frac{\delta}{d}\Phi X^T \left( I_n + \frac{\delta}{d}X\Phi X^T \right)^{-1}y.$$

*Proof.* Preconditioned gradient descent (23) is equivalent to mirror descent

$$\nabla\phi(w_{t+1}) = \nabla\phi(w_t) - \eta_t \nabla R(w_t)$$

with mirror map $\phi(w) = \frac{1}{2}w^T\Sigma w$. By a result of (Gunasekar et al., 2018), if mirror descent with mirror map $\phi$, a unique root loss function (e.g. the squared error loss), initialisation $w_0$ and stepsize $(\eta_t)_{t\in\mathbb{N}}$ satisfies $\lim_{t\to\infty} R(w_t) = 0$ then

$$\lim_{t\to\infty} w_t = \arg\min_{w\in\mathcal{G}} D_\phi(w, w_0),$$

where

$$D_\phi(w, w_0) = \phi(w) - \phi(w_0) - \nabla\phi(w_0)^T(w - w_0)$$

is the associated Bregman divergence. By this result applied with $\phi(w) = \frac{1}{2}w^T\Sigma w$, we have that if preconditioned gradient descent (23) initialized at $w_0$ converges, its limit satisfies

$$\lim_{t\to\infty} w_t = \arg\min_{w\in\mathbb{R}^d\,:\,Xw=y} \|\Sigma^{\frac{1}{2}}(w - w_0)\|_2^2.$$

After a linear transformation and an application of a result about approximate solutions to linear matrix equations (Penrose, 1956), similarly as in (4), we obtain

$$\lim_{t\to\infty} w_t = \Sigma^{-\frac{1}{2}}(X\Sigma^{-\frac{1}{2}})^{\dagger}(y - Xw_0) + w_0. \tag{24}$$

Finally, using

$$w_0 = \frac{\delta}{d}\Phi X^T\left(I_n + \frac{\delta}{d}X\Phi X^T\right)^{-1}y,$$

we obtain

$$\Sigma^{-\frac{1}{2}}(X\Sigma^{-\frac{1}{2}})^{\dagger}(y - Xw_0) + w_0 = \left(\frac{\delta}{d}\Phi X^T + \Sigma^{-\frac{1}{2}}(X\Sigma^{-\frac{1}{2}})^{\dagger}\right)\left(I_n + \frac{\delta}{d}X\Phi X^T\right)^{-1}y = w_O.$$

$\square$

## A.3 Proof of Proposition 3

We prove that for any deterministic initialization $w_0 \in \mathbb{R}^d$, the limit of converging preconditioned gradient descent $w_{t+1} = w_t - \eta_t\Sigma^{-1}\nabla R(w_t)$ satisfies that

$$\lim_{t\to\infty} w_t = \arg\min_{w\in\mathcal{G}} V(w).$$

*Proof.* Recall from (24) that

$$\lim_{t\to\infty} w_t = \Sigma^{-\frac{1}{2}}(X\Sigma^{-\frac{1}{2}})^{\dagger}(y - Xw_0) + w_0,$$

and the definition of the variance $V(w) = \mathbb{E}_{\xi,w^\star}\|w - \mathbb{E}(w|w^\star, X)\|_{\Sigma}^2$. Therefore, we have

$$\lim_{t\to\infty} w_t - \mathbb{E}(\lim_{t\to\infty} w_t|w^\star, X) = \Sigma^{-\frac{1}{2}}(X\Sigma^{-\frac{1}{2}})^{\dagger}\xi.$$

Moreover, the optimal interpolator of Definition 1 satisfies

$$w_b = w^\star + \Sigma^{-\frac{1}{2}}(X\Sigma^{-\frac{1}{2}})^{\dagger}\xi,$$

so that

$$w_b - \mathbb{E}(w_b|w^\star, X) = \Sigma^{-\frac{1}{2}}(X\Sigma^{-\frac{1}{2}})^{\dagger}\xi$$

and hence

$$V(w_b) = V(\lim_{t\to\infty} w_t) = \mathbb{E}_{\xi}\|\Sigma^{-\frac{1}{2}}(X\Sigma^{-\frac{1}{2}})^{\dagger}\xi\|_{\Sigma}^2.$$

In other words, $\lim_{t\to\infty} w_t$ fits the noise in exactly the same way as the optimal interpolator $w_b$, which has the smallest possible risk among all interpolators. Hence, it is enough to show that $w_b$ also has smallest possible variance among all interpolators. We argue by contradiction. Assume that $\widehat{w}$ is an interpolator with smaller variance than $w_b$. Then

$$X\widehat{w} = y$$

implies that

$$X\big(\widehat{w} - \mathbb{E}(\widehat{w}|w^\star, X)\big) = \xi$$

and hence $w^\star + \widehat{w} - \mathbb{E}(\widehat{w}|w^\star, X)$ is also an interpolator. But $w^\star + \widehat{w} - \mathbb{E}(\widehat{w}|w^\star, X)$ has zero bias (recall that the definition of bias is $B(w) = \mathbb{E}_{\xi,w^\star}\|\mathbb{E}(w|w^\star, X) - w^\star\|_{\Sigma}^2$) and therefore, by assumption, has smaller risk than $w_b$. This is a contradiction. $\square$

## A.4 Assumption 1 implies rank$(X) = n$ with probability 1.

Note that

$$\begin{aligned}
\mathbb{P}(\text{rank}(X) \neq n) &= \mathbb{P}(|\text{Span}(x_1,\ldots,x_n)| < n)\\
&= \mathbb{P}(\cup_{i\in\{1,\ldots,n\}}\{x_i \in \text{Span}(x_1,\ldots,x_{i-1},x_{i+1},\ldots,x_n)\})\\
&\leq \sum_{i=1}^{n}\mathbb{P}(\{x_i \in \text{Span}(x_1,\ldots,x_{i-1},x_{i+1},\ldots,x_n)\})\\
&= \sum_{i=1}^{n}\mathbb{E}\big(\mathbb{P}(\{x_i \in \text{Span}(x_1,\ldots,x_{i-1},x_{i+1},\ldots,x_n)\}|x_1,\ldots,x_{i-1},x_{i+1},\ldots,x_n)\big)\\
&= 0,
\end{aligned}$$

where the last equation follows directly by applying Assumption 1.

## A.5 Response-linear interpolator with optimal bias

By choosing $\sigma^2 = 0$ in A.1, the proof of Proposition 1, one obtains the interpolator with optimal bias among response-linear achievable interpolators. This interpolator is

$$\Phi X^T (X \Phi X^T)^{-1} y,$$

which is in agreement with the asymptotic result of (Amari et al., 2021). Therefore, when the prior is isotropic, as claimed in 5.3 the interpolator with optimal bias among response-linear achievable interpolators is the minimum-norm interpolator.

## A.6 Proof of equation (14)

We prove that, when the prior is isotropic $\Phi = I_d$, under Assumptions 3 and 5, if $n, d \to \infty$ with $\frac{d}{n} \to \gamma > 1$ then we have with probability 1 that

$$\lim_{d \to \infty} \mathbb{E}_{\xi, w^*} r(w_V) - r(w^\star) = B(w_V) + V(w_V),$$

where

$$B(w_V) = r^2 \frac{\gamma - 1}{\gamma} \int_0^\infty s \, d\mathcal{H}(s),$$

$$V(w_V) = \frac{\sigma^2}{\gamma - 1}.$$

*Proof.* The proof uses techniques which were already developed in Hastie et al. (2019). Namely Theorem 1 of Rubio and Mestre (2011) and an exchange of limits. Recall that

$$w_V = \Sigma^{-\frac{1}{2}} (X\Sigma^{-\frac{1}{2}})^\dagger X w^\star + \Sigma^{-\frac{1}{2}} (X\Sigma^{-\frac{1}{2}})^\dagger \xi$$

and $Z = X\Sigma^{-\frac{1}{2}}$. Therefore, we have

$$\mathbb{E}_{\xi, w^\star} r(w_V) - r(w^\star) = B(w_V) + V(w_V),$$

where, it was proved in Hastie et al. (2019) that

$$V(w_V) = \sigma^2 \text{Tr}\big((Z^T Z)^\dagger\big) \longrightarrow \frac{\sigma^2}{\gamma - 1}$$

because $Z_i \overset{\text{i.i.d}}{\sim} \mathcal{N}(0, I_d)$. For the bias term we also use techniques similar to (Hastie et al., 2019). In particular, we have

$$B(w_V) = \mathbb{E}_{w^\star} w^{\star T} \Sigma^{\frac{1}{2}} (I - Z^\dagger Z)^T (I - Z^\dagger Z) \Sigma^{\frac{1}{2}} w^\star$$

$$= \mathbb{E}_{w^\star} \text{Tr}\big(\Sigma^{\frac{1}{2}} w^\star w^{\star T} \Sigma^{\frac{1}{2}} (I - Z^\dagger Z)\big)$$

$$= \frac{r^2}{d} \text{Tr}\big(\Sigma (I - Z^\dagger Z)\big).$$

Moreover, we have

$$Z^\dagger = (Z^T Z)^\dagger Z^T,$$

so that if we denote $\widehat{\Sigma} = Z^T Z / n$ to be the empirical covariance matrix of the whitened features, then

$$Z^\dagger Z = \widehat{\Sigma}^\dagger \widehat{\Sigma} = \lim_{\lambda \to 0^+} (\widehat{\Sigma} + \lambda I_d)^{-1} \widehat{\Sigma}.$$

Therefore,

$$B(w_V) = \lim_{\lambda \to 0^+} \frac{r^2}{d} \text{Tr}\big((I - (\widehat{\Sigma} + \lambda I_d)^{-1} \widehat{\Sigma}) \Sigma\big)$$

$$= \lim_{\lambda \to 0^+} \frac{r^2}{d} \lambda \text{Tr}\big((\widehat{\Sigma} + \lambda I_d)^{-1} \Sigma\big).$$

Now, we use Theorem 1 of (Rubio and Mestre, 2011) to compute the limit of $\mathrm{Tr}\big((\widehat{\Sigma} + \lambda I_d)^{-1}\Sigma\big)$ as $d/n \to \gamma > 1$ with $d \to \infty, n \to \infty$. This theorem shows that if $\Theta = (\Theta_d)_{d\in\mathbb{N}}$ is a sequence of matrices such that $\sqrt{\mathrm{Tr}\big(\Theta\Theta^T\big)}$ is uniformly bounded, then

$$\mathrm{Tr}\big(\Theta((\widehat{\Sigma} + \lambda I_d)^{-1} - c_d(\lambda)I_d)\big) \longrightarrow 0, \tag{25}$$

where $c_d(\lambda)$ is a certain quantity defined through an implicit equation (for simplicity we do not define it, as we only need to know its limit). If we choose $\Theta = I_d/d$, then because $\widehat{\Sigma} = Z^T Z/n$ where $Z_i \overset{\text{i.i.d}}{\sim} \mathcal{N}(0, I_d)$ and the spectral distribution $\mathcal{F}_{I_d}$ is just the distribution induced by the measure $\delta_1$ for all $d \in \mathbb{N}$, we have

$$\lim_{d\to\infty} c_d(\lambda) \to m(-\lambda),$$

where $m$ is the Stieltjes transform of the limiting spectral distribution of $Z^T Z/n$ given by the Marčenko-Pastur theorem (Marčenko and Pastur, 1967). Now that we know $c_d(\lambda) \to m(-\lambda)$, we use (25) again but with $\Theta = \Sigma/d$. This shows that

$$\mathrm{Tr}\left(\frac{\Sigma}{d}(\widehat{\Sigma} + \lambda I_d)^{-1}\right) - \mathrm{Tr}\left(\frac{\Sigma}{d}\right)m(-\lambda) \longrightarrow 0,$$

provided that $\sqrt{\mathrm{Tr}(\Sigma^2)}/d$ is uniformly bounded. This is true when Assumption 3 holds so that $\lambda_{\max}(\Sigma)$ is uniformly bounded. Moreover,

$$\frac{1}{d}\mathrm{Tr}(\Sigma) = \frac{1}{d}\sum_{i=1}^{d}\lambda_i(\Sigma) = \int s\, d\mathcal{F}_\Sigma(s) \longrightarrow \int s\, d\mathcal{H}(s),$$

where in the last line we used Assumptions 5 and 3. Therefore, we arrive at

$$\frac{r^2}{d}\lambda\mathrm{Tr}\big(\Sigma(\widehat{\Sigma} + \lambda I_d)^{-1}\big) \longrightarrow r^2\lambda m(-\lambda)\int s\, d\mathcal{H}(s).$$

Finally, assuming we can exchange limits (which we justify shortly), we have

$$\begin{aligned}
\lim_{d\to\infty} B(w_V) &= \lim_{d\to\infty}\lim_{\lambda\to 0^+}\frac{r^2}{d}\lambda\mathrm{Tr}\big(\Sigma(\widehat{\Sigma} + \lambda I_d)^{-1}\big) \\
&= \lim_{\lambda\to 0^+}\lim_{d\to\infty}\frac{r^2}{d}\lambda\mathrm{Tr}\big(\Sigma(\widehat{\Sigma} + \lambda I_d)^{-1}\big) \\
&= \lim_{\lambda\to 0^+} r^2\lambda m(-\lambda)\int s\, d\mathcal{H}(s),
\end{aligned} \tag{26}$$

and because $m$ is the Stieltjes transform of the standard Marčenko-Pastur law, it is known (Proposition 3.11 of Bai and Silverstein (2010)) that

$$\lim_{\lambda\to 0^+}\lambda m(-\lambda) = \frac{\gamma - 1}{\gamma}.$$

However, to fully finish the proof, one needs to first justify exchanging the limits in (26). We do this now. Define a sequence of functions $f_d : \mathbb{R}^+ \to \mathbb{R}$ with

$$f_d(\lambda) = \frac{r^2}{d}\lambda\mathrm{Tr}\big(\Sigma(\widehat{\Sigma} + \lambda I_d)^{-1}\big)$$

and

$$f(\lambda) = r^2\lambda m(-\lambda)\int s\, d\mathcal{H}(s).$$

We proved that $\lim_{d\to\infty} f_d(\lambda) = f(\lambda)$ pointwise. To assert

$$\lim_{d\to\infty}\lim_{\lambda\to 0^+} f_d(\lambda) = \lim_{\lambda\to 0^+} f(\lambda)$$

it is therefore, by the Moore-Osgood theorem, enough to show that $(f_d)_{d\in\mathbb{N}}$ is uniformly convergent. As $(f_d)_{d\in\mathbb{N}}$ has a pointwise limit, it is enough to show that every subsequence of $(f_d)_{d\in\mathbb{N}}$ has

a uniformly convergent subsequence. For this, we show that $(f_d)_{d\in\mathbb{N}}$ is uniformly bounded and has uniformly bounded derivative, which gives the convergent subsequences by the Arzela-Ascoli theorem. Indeed, we have that

$$|f_d(\lambda)| \le r^2 \lambda_{\max}(\Sigma)$$

and as $f_d'(\lambda) = \frac{r^2}{d}\mathrm{Tr}\big(\Sigma(\widehat{\Sigma} + \lambda I_d)^{-2}\widehat{\Sigma}\big)$ we have

$$|f_d'(\lambda)| \le r^2 \lambda_{\max}(\Sigma)\frac{\lambda_{\max}(\widehat{\Sigma})}{(\lambda_{\min}(\widehat{\Sigma})^+ + \lambda)^2} \le r^2 \lambda_{\max}(\Sigma)8\frac{(\sqrt{\gamma}+1)^2}{(\sqrt{\gamma}-1)^4}.$$

In the inequality we used Theorem 1 of Bai and Yin (1993) which shows that, with probability 1,

$$\liminf_{d\to\infty} \lambda_{\min}(\widehat{\Sigma})^+ \ge \frac{1}{2}(\sqrt{\gamma}-1)^2,$$

$$\limsup_{d\to\infty} \lambda_{\max}(\widehat{\Sigma}) \le 2(\sqrt{\gamma}+1)^2.$$

$\square$

## A.7 $V_{\ell_2}$ in strong weak features model with $\gamma\psi_1 = 1$.

In this subsection we justify the statement (of the last paragraph of Section 5.3) that, in the strong weak features model of covariance matrices

$$\Sigma = \mathrm{diag}(\rho_1,\ldots,\rho_1,\rho_2,\ldots,\rho_2) \in \mathbb{R}^{d\times d},$$

where the number of $\rho_1$s is $d \cdot \psi_1$ with $\psi_1 \in [0,1]$, we have

$$V_{\ell_2} = \sigma^2\left(\frac{v'(0)}{v(0)^2} - 1\right) \to \infty$$

as $\rho_2 \to 0$ for any $\gamma > 1$ such that $\gamma\psi_1 = 1$. Indeed, using the relation

$$m(z) + \frac{1}{z} = \gamma(v(z) + \frac{1}{z}),$$

(which can be shown to hold) and Definition 6 of the Stieltjes transform and its limit, it can be checked that

$$v(0) = \frac{x + \sqrt{x^2 + 4(\gamma-1)\rho_1\rho_2}}{2(\gamma-1)\rho_1\rho_2},$$

where $x = \rho_1 + \rho_2 - \gamma\psi_1\rho_1 - \gamma(1-\psi_1)\rho_2$. Moreover, taking a derivative in the Silverstein equation (Silverstein, 1995), which states that

$$-\frac{1}{v(z)} = z - \gamma\int\frac{s}{1 + sv(z)}\,d\mathcal{H}(s),$$

gives

$$\frac{v'(0)}{v(0)^2} - 1 = \gamma v'(0)\Delta, \tag{27}$$

where

$$\Delta = \left(\frac{\psi_1\rho_1^2}{(1+\rho_1 v(0))^2} + \frac{(1-\psi_1)\rho_2^2}{(1+\rho_2 v(0))^2}\right). \tag{28}$$

By rearranging (27) we obtain

$$\frac{v'(0)}{v(0)^2} - 1 = \frac{1}{1 - \gamma\Delta v(0)^2} - 1. \tag{29}$$

Now if $\gamma\psi_1 = 1$, then $x = \rho_2(2-\gamma)$ and

$$v(0) = \frac{2-\gamma + \sqrt{(2-\gamma)^2 + 4(\gamma-1)\frac{\rho_1}{\rho_2}}}{2(\gamma-1)\rho_1}.$$

Hence

$$v(0)\sqrt{\rho_2} \longrightarrow \sqrt{\frac{1}{(\gamma-1)\rho_1}}$$

as $\rho_2 \to 0$. Using (28) and (29), it can be therefore checked that, as $\rho_2 \to 0$,

$$\sqrt{\rho_2}\left(\frac{v'(0)}{v(0)^2}-1\right) \longrightarrow \frac{1}{2}\sqrt{\frac{\rho_1}{\gamma-1}}.$$

Therefore, $\frac{v'(0)}{v(0)^2} - 1 \to \infty$ as $\rho_2 \to 0$.

## A.8 Empirical comparison of the Graphical Lasso for some covariance matrices

We illustrate how the interpolator $w_{Oe}$, obtained by using the Graphical Lasso approximation of the covariance matrix, performs in comparison to the optimal response-linear achievable interpolator $w_O$ for two regimes of covariance matrices. In this Section, we do this in the regime of an isotropic prior $\Phi = I_d$. See Section A.9 for the case $\Phi \neq I_d$. The interpolator

$$w_{Oe} = \left(\frac{\delta_e}{d}X^T + \Sigma_e^{-\frac{1}{2}}(X\Sigma_e^{-\frac{1}{2}})^\dagger\right)\left(I_n + \frac{\delta_e}{d}XX^T\right)^{-1}y,$$

is constructed by using the Graphical Lasso estimator $\Sigma_e$ (Friedman et al., 2007) of the covariance matrix (implemented in scikit-learn (Pedregosa et al., 2011)), and choosing $\delta_e$ which minimizes the crossvalidated error on random subsets of the data as described in Section 5.1.

First, we look at the autoregressive regime, where

$$\Sigma_{i,j} = \rho^{|i-j|}$$

for all $i, j \in \{1, \ldots, d\}$ and $\rho \in (0, 1)$.

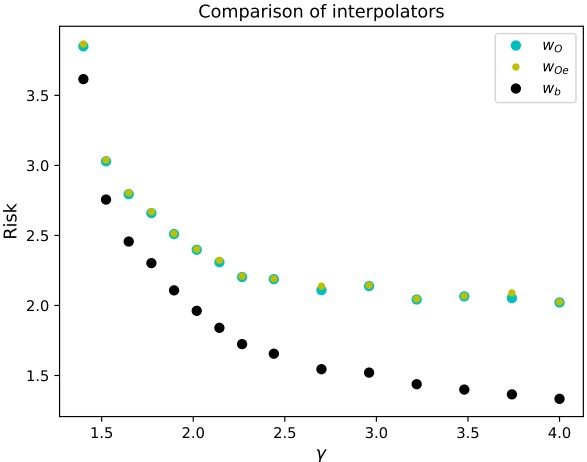

Figure 3: Plot of $\mathbb{E}_\xi r(w)$ (points) for $w \in \{w_O, w_{Oe}, w_b\}$ in the autoregressive regime with $d = \lfloor \gamma n \rfloor, r^2 = 1, \sigma^2 = 1, n = 2000, \rho = 0.5$.

Second, we consider an exponential regime (Dobriban and Wager, 2015), where the eigenvalues of $\Sigma$ are evenly spaced quantiles of the standard exponential distribution. Namely,

$$\Sigma_{i,i} = -\log(1 - p_i),$$

where $p_i = i/(d+1) \in (0, 1)$ for $i \in \{1, \ldots, d\}$. The off-diagonal entries are 0.

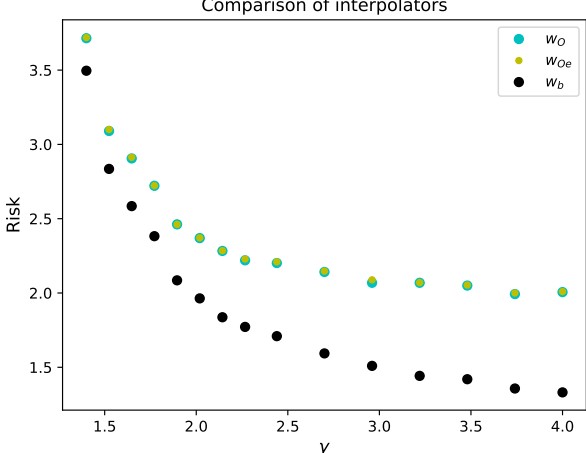

Figure 4: Plot of $\mathbb{E}_\xi r(w)$ (points) for $w \in \{w_O, w_{Oe}, w_b\}$ in the exponential regime with $d = \lfloor \gamma n \rfloor, r^2 = 1, \sigma^2 = 1, n = 2000$.

Note that the Graphical Lasso works well in the regimes of covariance matrices that we presented, because in these regimes the empirical-covariance-based estimator $w_{Oe}$ is seen to reproduce the behaviour of the population-covariance-based estimator $w_O$.

However, we do not make the claim that the Graphical Lasso approximation will approximate the population covariance matrix well in general. The covariance matrices considered in this work have a notable sparsity structure, and the Graphical Lasso approximation may not perform well for dense covariance matrices.

It is interesting to study which covariance matrix approximators one should use. If we consider $\Sigma_e = X^T X / n + \lambda I_d$ for any $\lambda \in \mathbb{R}$, one can check using the singular value decomposition of $X$ that

$$\Sigma_e^{-\frac{1}{2}} (X \Sigma_e^{-\frac{1}{2}})^\dagger y = X^\dagger y,$$

so that the corresponding preconditioned gradient descent converges to the same limit as gradient descent and hence removes the benefit of preconditioning. The last statement is also true when using the Ledoit-Wolf shrinkage covariance approximation (Ledoit and Wolf, 2004).

### A.9 Empirical approximation in non-isotropic regimes

In the examples considered so far, we empirically illustrated that $w_{Oe}$ approximates $w_O$ well. However, the considered examples used an isotropic prior, i.e. $\Phi = I_d$. It is natural to ask whether we are also able to match the generalization performance of $w_O$ when $\Phi \neq I_d$.

If we knew $\Phi$, or had some prior information about $\Phi$, then we can incorporate this information into an estimate $\widehat{\Phi}$ and use the fully empirical approximation

$$w_{Oe\widehat{\Phi}} = \left( \frac{\delta_e}{d} \widehat{\Phi} X^T + \Sigma_e^{-\frac{1}{2}} (X \Sigma_e^{-\frac{1}{2}})^\dagger \right) \left( I_n + \frac{\delta_e}{d} X \widehat{\Phi} X^T \right)^{-1} y, \tag{30}$$

which is likely to perform better than if we used $\widehat{\Phi} = I_d$ as in $w_{Oe}$ (11).

However, in Figures 5, 6 we empirically illustrate that the interpolator $w_{Oe}$ has generalization very similar to that of $w_O$ and $w_{Oe\Phi}$ even in regimes when the prior is not isotropic ($\Phi \neq I_d$). Using $w_{Oe}$ corresponds to having no information about the prior, while $w_{Oe\Phi}$ corresponds to having complete information about the covariance matrix of the prior. We see that both $w_{Oe\Phi}$ and $w_{Oe}$ approximate $w_O$ well in terms of generalization performance.

In Figure 5, we consider a prior where $\Phi$ is in the autoregressive regime. That is

$$\Phi_{ij} = \rho^{|i-j|}$$

for all $i, j \in \{1, \dots, d\}$ and we set $\rho = 0.5$. The population covariance matrix $\Sigma$ is in the exponential regime (Dobriban and Wager, 2015), where the eigenvalues of $\Sigma$ are evenly spaced quantiles of the standard exponential distribution. Namely,

$$\Sigma_{ii} = -\log\big(1 - i/(d+1)\big),$$

and the off-diagonal entries are 0. In Figure 6 we set $\Sigma$ to be in the autoregressive regime with $\rho = 0.5$ and we consider the "hard prior" regime (Richards et al., 2021) where $\Phi = \Sigma^{-1}$.

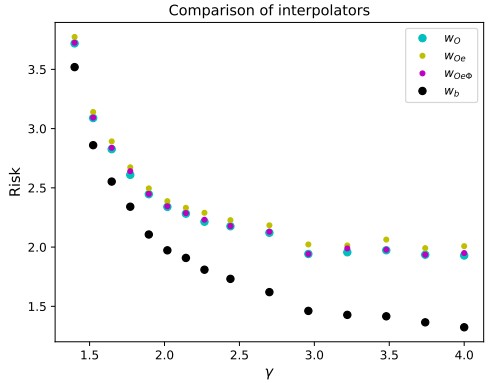
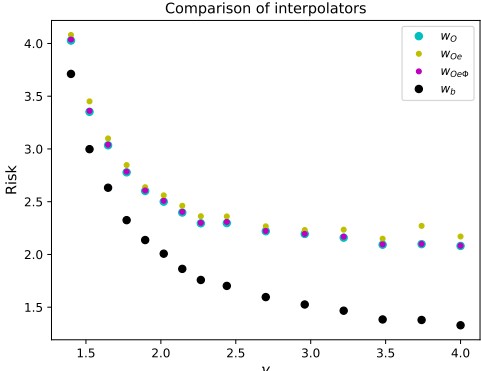

Figure 5: Plot of $\mathbb{E}_\xi r(w)$ for $w \in \{w_O, w_{Oe}, w_{Oe\Phi}, w_b\}$ with $r^2 = 1, \sigma^2 = 1, \gamma = \lfloor d/n \rfloor, n = 2000$. $\Sigma$ follows the exponential regime and $\Phi$ follows the autoregressive regime with $\rho = 0.5$.

Figure 6: Plot of $\mathbb{E}_\xi r(w)$ for $w \in \{w_O, w_{Oe}, w_{Oe\Phi}, w_b\}$ with $r^2 = 1, \sigma^2 = 1, \gamma = \lfloor d/n \rfloor, n = 2000$. $\Sigma$ follows the autoregressive regime with $\rho = 0.5$ and $\Phi = \Sigma^{-1}$ follows the "hard" prior regime.

### A.10  Proof of Propostion 4

We prove that the optimal response-linear achievable interpolator in random features regression is

$$a_O = \Sigma_z^{-1}\left(\Sigma_{zx}\Phi X^T + Z^T\big(Z\Sigma_z^{-1}Z^T\big)^{-1}\big(\frac{d}{\delta}I_n + X\Phi X^T - Z\Sigma_z^{-1}\Sigma_{zx}\Phi X^T\big)\right)\left(\frac{d}{\delta}I_n + X\Phi X^T\right)^{-1}y.$$

The proof follows analogous steps to the proof of Proposition 1.

*Proof.* First, we note that as $a_O \in \mathcal{L}$, there exists $\mathbb{R}^{d \times n} \ni Q = Q(X, \Sigma, \Phi, \delta)$ such that $a_O = Qy = QXw^\star + Q\xi$. Therefore, the definition of $a_O$,

$$a_O = \underset{a \in \mathcal{G} \cap \mathcal{L}}{\arg\min} \mathbb{E}_{\xi, w^\star} r(f_a) - r(w^\star),$$

can be restated as

$$a_O = \underset{\substack{a_O = Qy \\ ZQy = y}}{\arg\min} \mathbb{E}_{\xi, w^\star} f_1(Q) + f_2(Q) + f_3(Q), \tag{31}$$

where

$$f_1(Q) = \frac{r^2}{d}\mathrm{Tr}\big(\Sigma_z Q X \Phi X^T Q^T\big)$$
$$f_2(Q) = \sigma^2 \mathrm{Tr}\big(\Sigma_z Q Q^T\big)$$
$$f_3(Q) = -2\frac{r^2}{d}\mathrm{Tr}\big(Q^T \Sigma_{zx}\Phi X^T\big),$$

where $\Sigma_z = \mathbb{E}_{\tilde{x}}(\sigma(\Theta\tilde{x}/\sqrt{d})\sigma(\Theta\tilde{x}/\sqrt{d})^T)$ and $\Sigma_{zx} = \mathbb{E}_{\tilde{x}}(\sigma(\Theta\tilde{x}/\sqrt{d})\tilde{x}^T)$. Moreover, $ZQy = y$ almost surely implies that $ZQX = X$ almost surely. This is because taking expectation with respect to $\xi$ in $ZQy = y$ implies

$$(ZQX - X)w^\star = 0$$

and therefore
$$0 = \mathbb{E}_{w^\star}\big(\|(ZQX - X)w^\star\|_2^2 \mid w^\star\big) = w^{\star T}\mathbb{E}\big((ZQX - X)^T(ZQX - X)\big)w^\star.$$

Because this holds almost surely for all realizations of $w^\star \in \mathbb{R}^d$, similarly as in Section A.1, it follows that $\mathbb{E}\big((ZQX - X)^T(ZQX - X)\big) = 0$. Finally, therefore also $\mathbb{E}\big(\text{Tr}((ZQX - X)^T(ZQX - X))\big) = 0$ and because $\|A\| = \sqrt{\text{Tr}(AA^T)}$ is a norm, this implies that $ZQX - X = 0$ almost surely. Hence, (31) is equivalent to

$$w_O = \underset{\substack{a_O = Qy \\ ZQX = X}}{\arg\min}\ f_1(Q) + f_2(Q) + f_3(Q). \tag{32}$$

Now we use Theorem 2 of Penrose (1955) which states that for any matrices $A, B, C$ and $D$, all solutions $B$ to the equation $ABC = D$ can be written as $B = A^\dagger DC^\dagger + S - A^\dagger ASCC^\dagger$ where $S$ is arbitrary. Therefore, $ZQX = X$ is equivalent to

$$Q = Z^\dagger XX^\dagger + S - Z^\dagger ZSXX^\dagger = Z^\dagger + S - Z^\dagger ZS,$$

for some arbitrary $S \in \mathbb{R}^{N \times n}$. Hence, (32) is equivalent to an unconstrained optimization problem over $\mathbb{R}^{N \times n}$ in the form

$$w_O = \underset{S}{\arg\min}\ f(S), \tag{33}$$

where $f(S) = f_1(a_O(S)) + f_2(a_O(S)) + f_3(a_O(S))$. Now we show that $f : \mathbb{R}^{N \times n} \to \mathbb{R}$ is strictly convex. This is done in precisely the same way as in the proof A.1. Namely, we note that $\|A\| = \sqrt{\text{Tr}(AA^T)}$ is a norm and hence $\mathbb{R}^{N \times N} \ni A \mapsto \text{Tr}(AA^T)$ is strictly convex. As $f_1(S), f_2(S)$ are compositions of an affine map with $\mathbb{R}^{N \times N} \ni A \mapsto \text{Tr}(AA^T)$, and by noting that $f_3(S)$ is affine, it follows that $S \mapsto f(S)$ is strictly convex. Moreover, $f$ is differentiable. Therefore, to find a unique global minimum of $f$, it is enough to find $S^\star \in \mathbb{R}^{N \times n}$ such that $\partial f(S^\star) = 0$. Using tools of matrix calculus we find

$$\partial f(S) = 2(I_N - Z^\dagger Z)\Sigma A,$$

where

$$A = \left(\sigma^2 S + \frac{r^2}{d}\big(SX - \Sigma_z^{-1}\Sigma_{zx}\big)\Phi X^T + \Sigma_z Z^\dagger\big(I_n - ZS\big)\big(\sigma^2 I_n + \frac{r^2}{d}X\Phi X^T\big)\right).$$

Because $\mathbb{R}^N \ni v \mapsto (I_N - Z^\dagger Z)v$ is the projection onto $\text{Ker}(Z) = \text{Im}(Z^T)^\perp$, this hints towards finding $S^\star$ such that $A = \Sigma^{-1}Z^T B$ for some matrix $B$. This is achieved, for example, if

$$\sigma^2 S^\star + \frac{r^2}{d}\big(S^\star X - \Sigma_z^{-1}\Sigma_{zx}\big)\Phi X^T = \Sigma^{-1}Z^T B$$

and

$$I_n - ZS^\star = 0,$$

for some matrix $B$. The first equation implies

$$S^\star = \Sigma_z^{-1}\left(\frac{r^2}{d}\Sigma_{zx}\Phi X^T + Z^T B\right)\left(\sigma^2 I_n + \frac{r^2}{d}X\Phi X^T\right)^{-1} \tag{34}$$

and using that $ZS^\star = I_n$ gives

$$B = \left(Z\Sigma_z^{-1}Z^T\right)^{-1}\left(\sigma^2 I_n + \frac{r^2}{d}X\Phi X^T - \frac{r^2}{d}Z\Sigma_z^{-1}\Sigma_{zx}\Phi X^T\right).$$

Plugging $B$ back into (34) gives

$$S^\star = \Sigma_z^{-1}\left(\Sigma_{zx}\Phi X^T + Z^T\big(Z\Sigma_z^{-1}Z^T\big)^{-1}\big(\frac{d}{\delta}I_n + X\Phi X^T - Z\Sigma_z^{-1}\Sigma_{zx}\Phi X^T\big)\right)\left(\frac{d}{\delta}I_n + X\Phi X^T\right)^{-1}.$$

Finaly, because $ZS^\star = I_n$, it follows that $Q^\star = Z^\dagger + S^\star - Z^\dagger ZS^\star = S^\star$ and hence

$$a_O = \Sigma_z^{-1}\left(\Sigma_{zx}\Phi X^T + Z^T\big(Z\Sigma_z^{-1}Z^T\big)^{-1}\big(\frac{d}{\delta}I_n + X\Phi X^T - Z\Sigma_z^{-1}\Sigma_{zx}\Phi X^T\big)\right)\left(\frac{d}{\delta}I_n + X\Phi X^T\right)^{-1}y.$$

$\square$

## A.11 Proof of Proposition 5

We prove that the optimal response-linear achievable interpolator $a_O$ in random features regression is the limit of preconditioned gradient descent on the last layer,

$$w_{t+1} = w_t - \eta_t \Sigma_z^{-1} \nabla R(w_t),$$

provided that the algorithm converges and initialized at

$$a_0 = \Sigma_z^{-1} \Sigma_{zx} \Phi X^T \left( \frac{d}{\delta} I_n + X \Phi X^T \right)^{-1} y.$$

*Proof.* As before, by a result of (Gunasekar et al., 2018) we have that the limit of

$$w_{t+1} = w_t - \eta_t \Sigma_z^{-1} \nabla R(w_t),$$

on the last layer, initialized at some $a_0$ and provided that it converges, satisfies

$$\lim_{t \to \infty} w_t = \Sigma_z^{-\frac{1}{2}} \left( Z \Sigma_z^{-\frac{1}{2}} \right)^\dagger \left( y - Z a_0 \right) + a_0.$$

Using

$$a_0 = \Sigma_z^{-1} \Sigma_{zx} \Phi X^T \left( \frac{d}{\delta} I_n + X \Phi X^T \right)^{-1} y,$$

we obtain

$$\Sigma_z^{-\frac{1}{2}} \left( Z \Sigma_z^{-\frac{1}{2}} \right)^\dagger \left( y - Z a_0 \right) + a_0 =$$

$$\Sigma_z^{-1} \left( \Sigma_{zx} \Phi X^T + Z^T \left( Z \Sigma_z^{-1} Z^T \right)^{-1} \left( \frac{d}{\delta} I_n + X \Phi X^T - Z \Sigma_z^{-1} \Sigma_{zx} \Phi X^T \right) \right) \left( \frac{d}{\delta} I_n + X \Phi X^T \right)^{-1} y.$$

$\square$

## A.12  Random features example

We illustrate the test error of the random features model $x \mapsto f_a(x) = a^T \sigma(\Theta x / \sqrt{d})$ for the optimal response-linear achievable interpolator, $f_{a_O}$, with

$$a_O = \Sigma_z^{-1} \left( \Sigma_{zx} \Phi X^T + Z^T \left( Z \Sigma_z^{-1} Z^T \right)^{-1} \left( \frac{d}{\delta} I_n + X \Phi X^T - Z \Sigma_z^{-1} \Sigma_{zx} \Phi X^T \right) \right) \left( \frac{d}{\delta} I_n + X \Phi X^T \right)^{-1} y$$

in comparison to the test error for the minimum-norm interpolator $a_{\ell_2} = Z^\dagger y$ on a standard example. Let $x_i \sim \text{Unif}(\mathbb{S}^{d-1}(\sqrt{d}))$ and $\Theta \in \mathbb{R}^{N \times d}$ be randomly initialized such that the rows of $\Theta$ satisfy $\Theta_i \in \mathbb{S}^{d-1}(\sqrt{d})$. Here, $\mathbb{S}^{d-1}(\sqrt{d})$ is the sphere with radius $\sqrt{d}$ in $\mathbb{R}^d$. We numerically compute $\Sigma_z = \mathbb{E}_{\tilde{x}}(\sigma(\Theta \tilde{x}/\sqrt{d})\sigma(\Theta \tilde{x}/\sqrt{d})^T)$ and $\Sigma_{zx} = \mathbb{E}_{\tilde{x}}(\sigma(\Theta \tilde{x}/\sqrt{d})\tilde{x}^T)$ by sampling from $\text{Unif}(\mathbb{S}^{d-1}(\sqrt{d}))$ and use the true signal-to-noise ratio $\delta$. We observe, as expected, that $f_{a_O}$ generalizes better than $f_{a_{\ell_2}}$. This is so even for large $\gamma = \lfloor N/d \rfloor$, where Mei and Montanari (2019) showed that, for high-enough signal-to-noise ratio, the test error of the minimum-norm interpolator converges to the test error of the optimally-tuned ridge regression estimator in the limit as $N/d \to \infty$ (under certain assumptions).

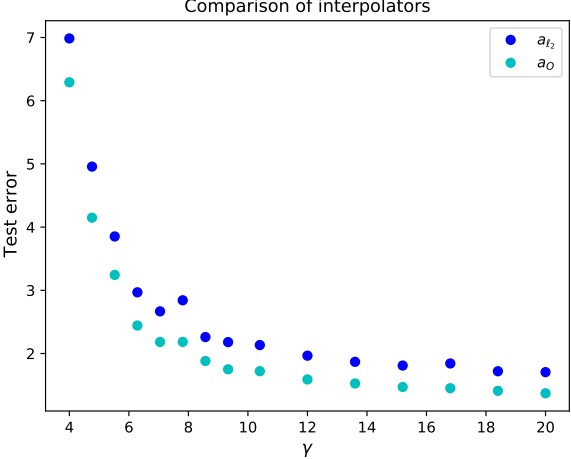

Figure 7: Plot of test error of $f \in \{f_{a_{\ell_2}}, f_{a_O}\}$ for $\gamma = \lfloor N/d \rfloor$ when $x_i \sim \text{Unif}(\mathbb{S}^{d-1}(\sqrt{d}))$ with $r^2 = 5, \sigma^2 = 1, \lfloor n/d \rfloor = 3, n = 2000$.