# OpenReview forum: "On Optimal Interpolation in Linear Regression"
_NeurIPS.cc/2021/Conference — NeurIPS 2021 Poster_

### Official Review · Reviewer_raJt · 2021-07-07

**Rating:** 4
**Confidence:** 3

**Summary:**

This paper considers the problem of optimal interpolation in linear regression. It defines the class of response-linear achievable interpolators, and provides a closed-form expression for the optimal interpolator in this class. The paper is about an important topic in machine learning and statistics. However, it is unclear to me whether the "achievable" interpolators are really achievable, and as a result if the investigation is meaningful.

**Limitations And Societal Impact:**

Please see the main review.

**Main Review:**

Major comments and questions:
1. The paper claims that the realization of true data-generating parameter and the realization of the noise in the training data are unknown. However, it assumes that the population covariance, the signal-to-noise ratio, and the covariance of the prior for the signal are known. The latter seems to be more unrealistic. For high dimensional data, the covariance matrix contains much more number of parameters than the regression coefficient so it is a much harder problem to estimate the covariance matrix. In a Bayesian setting, the primary goal is to find the distribution of the parameter. If the distribution of the parameter is assumed to be known, then there is no estimation problem.

2. For $\Phi$, do you mean it is the correlation matrix? Otherwise calling $r^2$ the signal is not appropriate.

3. The population risk between lines 125 and 126 is not clear.It is written as the conditional expectation given $X, \xi, w^*$, but this does not make sense. A risk should not be random.

4. I think the $y$ in line 131 should be $y_i$, but what is the $y$ in line 132? The $y$ in line 117 should be $y_i$? How about the $y$ in lines 120 and 124? Each of these may be typos or poor notations, but when there are too many this type of problems and makes it hard to following the paper it becomes a major issue.

5. Line 142, I believe you need $\gamma\ge1$.

6. Line 156 says "rank(X) = n with probability 1" while this becomes a condition in line 160.

7. Definition 4, do you mean there exists a known function? Otherwise, the function $f$ may depends further on other unknowns.

8. As in my first point, the $w_O$ is not realistic. The updating formula in (7) requires the inverse of $\Sigma$ which is almost impossible for high dimensional data.

9. I don't think Hastie et al. (2019) considered the case of $\Phi=I_d$. It is probably a typo here.


Minor issues and typos:

Line 53, "are best" to "are the best".

In (5), use paratheses to make the definition clear.


**Time Spent Reviewing:**

about four hours

---

### Official Review · Reviewer_hWGm · 2021-07-14

**Rating:** 6
**Confidence:** 3

**Summary:**

The paper under reviews studies the problem of optimal linear interpolation. More precisely the authors derive an explicit formula for the *best* linear interpolator that can be written as a linear function of the outputs and as a function of desired quantities. From this, they show how to recover theoretically this estimator from a optimization algorithm. Finally they expose cases where the designed estimator show some good generalization properties while the miminum norm interpolator does not.  The link with kernel methods is also made.

**Limitations And Societal Impact:**

Yes

**Main Review:**

**Overall comments and impression**

First, I would like to say that I am not an expert in statistical learning theory, so that I may have missed some insights in the results.

To begin, I would like to say that the paper is perfectly well written: the story is very clear and the mathematical writing is precise and fluid. This made the reading very pleasant and I would like to thank the authors for that.

The aim of the author is clear: provide an (under certain constraints) optimal linear predictor that may perform better than the minimum norm one naturally considered in the literature (naturally coming from GD or SGD on the empirical risk). The authors do so quite effortlessly ad give, to my opinion a nice result for those asking whether minimum norm solutions are always good estimators. **Yet**, I would have liked a lot more comments on the shape of this estimator: what intuition should the reader build when reading the paper ? A clear weakness is that the authors never really take the time of analysing why makes the estimator optimal (beyond technical considerations). In the very same manner, Proposition 2 gives that the optimal linear response is in fact the solution recovered by GD with a certain initialization: could the authors comment a bit on this ?

The result on the divergence of the minimum norm linear interpolator (whereas the optimal one does not) is nice: however I would really need to understand further *why* and *when* we should expect this to happen. And in general cases *how close* should I expect the minimum norm interpolator to be close to the optimal one ?


**Minor comments:**

- I guess that the part on the random matrix concepts arrive too early. As far as I understood, there is absolutely no need to put this part at the beginning, and this could be postpone in section 4.2 or even in the Appendix.

- The part on random features model gives nothing new to the results: they are not commented (at all) and are absolutely expected once the linear model is covered. If these results are not further commented, then I recommend to put it in the Appendix as they do not give any additional insight on the paper. A nice introduction on why linear model are trendy is enough in my opinion (NTK, random features...)

- In Section 4.2, is this possible to derive a close form for the asymptotic risk of the optimal interpolator ? If not, why is this complicated ?

- Give more motivation and justification for the form of the *optimal linear interpolating estimator* you build. Why are you okay with the dependance on \Sigma, \phi, \delta. How could relax these ? I understand that these parameter will be estimated on the fly later: but once again, the comments on this are not deep precise enough to convince me. In particular there are only a few comments on the Graphiical Lasso approximation and why this is a good idea to approximate the population covariance.

- Plots commenting Proposition 2 would rely complement the result! If not, it seems not so important: a comparison with GD without preconditionning would be welcomed.


**Conclusion**

Even if I really enjoyed reading the paper, my opinion is that, under this form, the results lack from commentaries, intuition, insights and a bit of further developments.

**Time Spent Reviewing:**

5

---

> ### Author Response · Authors · 2021-08-10
> **Response to reviewer hWGm**
>
> We thank the reviewer for appreciating the clarity of the exposition and are glad that the reviewer enjoyed reading the paper. We understand and appreciate their comment that they are not an expert in statistical learning theory. We thank them for a constructive review which will improve our paper and we provide clarifications which we hope to be helpful.
>
> ## Intuition and shape of the estimator
> The reviewer states:
>
> "I would have liked a lot more comments on the shape of this estimator: what intuition should the reader build when reading the paper ? A clear weakness is that the authors never really take the time of analysing why makes the estimator optimal (beyond technical considerations). "
>
> It is indeed not clear how to provide good intuition about the "shape" of the estimator. It is optimal *by definition* and, as the reviewer points out, it is obtained as a solution of an optimization problem. In the next section, we provide some intuition and motivation about the general investigation of this paper. Some intuition about the "shape" of the interpolator can be perhaps obtained from our answer in section Comments on Proposition 2.
>
> ## General motivation and intuition
>
> We thought it could be a good idea to summarize some of the high-level motivations for the investigation of the optimal response-linear interpolator in our work.
>
> Neural networks are often trained with gradient descent until interpolation. Gradient descent has a particular implicit bias (typically some solution with minimal $\ell_2$ norm). There is usually infinitely many interpolators, so why should it be that these gradient descent solutions are the best?
>
> Can we construct an interpolator that will provably achieve the best solution? This is what motivates our paper. However, it is hard to answer this question about neural networks, so we start with the simplest possible case - linear regression. Here, we first find what is the best possible solution that we could hope to get (the optimal response-linear achievable interpolator) and then find whether there is an empirical algorithm that could achieve it.
>
> From our results, this best interpolator is achieved by preconditioned gradient descent (and can't be achieved by gradient descent) with a very specific initialization. In fact, our experiments show that the solution achieved by 0-initialized gradient descent can perform arbitrarily worse than the best interpolator. Therefore, in our opinion, this poses the following intersting question: If the aforementioned phenomenon happens in models as simple as linear regression, then what makes us think that training neural networks with gradient descent is the best way?
>
> ## Comments on Proposition 2
> The reviewer asks:
>
> "In the very same manner, Proposition 2 gives that the optimal linear response is in fact the solution recovered by gradient descent with a certain initialization: could the authors comment a bit on this ?"
>
> This is not correct (typo?). Proposition 2 states that the optimal response-linear interpolator is recovered by gradient descent *preconditioned* with the inverse of the population covariance matrix of the data - *not* by gradient descent. In fact, it can be checked that it *can not* be recovered by gradient descent in general. The solution that is recovered by gradient descent initialized at $w_0$ takes the form
>
> $w=X^\dagger(y-Xw_0)+w_0,$
>
> while the solution that is recovered by gradient descent preconditioned with some matrix $P$ and initialized at $w_0$ takes the form
>
> $w=P^{\frac{1}{2}}(X P^{\frac{1}{2}})^\dagger(y-Xw_0)+w_0.$
>
> Proposition 2 states that the optimal repsponse-linear interpolator is achieved with preconditioned gradient descent when we choose $P = \Sigma^{-1}$ and $w_0=\frac{\delta}{d}\Phi X^{T}(I_n+\frac{\delta}{d}X\Phi X^T)^{-1}y$. This initialization $w_0$ can also be noticed to be a generalized ridge regressor. This is because it can be checked that
>
> $\frac{\delta}{d}\Phi X^{T}\bigg(I_n+\frac{\delta}{d}X\Phi X^T\bigg)^{-1}y=\bigg(X^TX+\frac{\delta}{d}\Phi\bigg)^{-1}X^Ty,&emsp;&emsp;(1)$
>
> which is the limit of gradient descent (initialized at $0$)  on the empirical risk with a generalized ridge penalty:
>
> $R(w)=\frac{1}{n}\sum_{i=1}^n (x_i^Tw-y_i)^2+\frac{\delta}{d}w^T\Phi w .$
>
> For the last statement, see Denny Wu and Ji Xu, NeurIPS 2020. Hence, another intuitive way of how to think about the optimal response-linear interpolator is to first perform gradient descent (initialized at $0$) on generalized ridge regression, and after convergence, perform gradient descent preconditioned with the inverse of the population matrix of the data (initialized at the output of the previous algorithm).
>
> ## Intuition on divergence in Figures 1 and 2
> The reviewer states:
>
> "The result on the divergence of the minimum norm linear interpolator (whereas the optimal one does not) is nice: however I would really need to understand further why and when we should expect this to happen."
>
> The divergence in Figure 1 is related to the double descent phenomenon. It is known that the generalization error of the minimum-norm solution diverges as $d/n\to 1$. In Figure 1, we have the covariance matrix $\Sigma = \text{diag}(\rho_1,\dots, \rho_1, \rho_2, \dots, \rho_2)$, where  $\rho_1=1$ and take $\rho_2\to 0$ while $d/n = 2$ and $\psi_1 = \frac{1}{2}$ (the fraction of $\rho_1$s in $\Sigma$). When $\rho_2 \approx 0$, the regression problem is (approximately) only concentrated on the first $\tilde{d}:=\psi_1d = d/2$ components of $w^{\star}$. Because $\tilde{d}/n = (d/2)/n = 1$, we observe the divergence of the minimum-norm solution analogous to the double descent phenomenon.
>
> This phenomenon does not happen for the optimal response-linear interpolator because the population covariance matrix (which causes the divergence) is used in its construction in a way such that its variance term is upper bounded by a quantity which is independent of the covariance matrix (this argument is made technically precise in Section 4.2). We hope this discussion, which we plan to include in the camera-ready version of our work, can address the reviewer's concern.
>
> ## Closeness of interpolators
> The reviewer asks:
>
> "And in general cases how close should I expect the minimum norm interpolator to be close to the optimal one ?"
>
> In general, we do *not* expect them to be close, as the example in Figure 3 in our paper illustrates.
>
> ## Asymptotic risk of optimal response-linear interpolator
> The reviewer asks:
>
> "In Section 4.2, is this possible to derive a close form for the asymptotic risk of the optimal interpolator ? If not, why is this complicated ?"
>
> We have investigated this. This is technically challenging. We illustrate why. The variance term of the optimal response-linear interpolator can be checked to be
>
> $V(w_{O})=\frac{\delta^2}{d^2}\text{Tr}\Bigg(X\Phi\Sigma\Phi X^T\bigg(I_n+\frac{\delta}{d}X\Phi X^T\bigg)^{-2}\Bigg)&emsp;&emsp; (2)$
>
> $+ \text{Tr}\Bigg({(X {\Sigma^{-\frac{1}{2}}})^\dagger}^T(X{\Sigma^{-\frac{1}{2}}})^\dagger\bigg(I_n+\frac{\delta}{d}X\Phi X^T\bigg)^{-2}\Bigg)&emsp;&emsp; (3)$
>
> $+ 2\frac{\delta}{d}\text{Tr}\Bigg({(X{\Sigma^{-\frac{1}{2}}})^\dagger}^T{\Sigma^{\frac{1}{2}}}\Phi X^T\bigg(I_n+\frac{\delta}{d}X\Phi X^T\bigg)^{-2}\Bigg)&emsp;&emsp; (4)$
>
> Assume for simplicity that $\Phi=I_d$ (we will show difficulty in calculating the asymptotic limit even in this case). The techniques to compute the limit of (2),(3),(4) often reduce to results analogous to Theorem 1 of Spectral Convergence for a General Class of Random Matrices by Rubio and Mestre, or the proof of Lemma 7.4 of High-Dimensional Asymptotics of Prediction: Ridge Regression and Classification by Dobriban and Wager, where limits of expressions of the form
>
> $\text{Tr}\bigg(\Theta(\lambda I_d+X^TX)^{-\alpha}\bigg)&emsp;&emsp;(5)$
>
> are computed (as $d/n\to\gamma$), for some constant $\lambda, \alpha\in\\{1,2\\}$ and a *deterministic* sequence of matrices $\Theta = (\Theta_d)_{d\in\mathbb{N}}$ with a uniformly bounded trace norm.
>
> The problem is that in (2),(3) and (4), the matrices which would represent $\Theta$ in (5) are random, and hence it is not clear how to use (5) .
>
> Sometimes it is possible to bypass this with a trick - e.g. (2) can be rewritten as in line 4, p. 22 of https://arxiv.org/abs/2006.06386 (Richards et al.) and hence its limit computed.
>
> However, it is not clear how to deal with terms (3) and (4), which poses the technical challenge.
>
> ## Intuition, motivation and dependence on parameters
> The reviewer states:
>
> "Give more motivation and justification for the form of the optimal linear interpolating estimator you build. Why are you okay with the dependance on $\Sigma, \phi, \delta$. How could relax these ? I understand that these parameter will be estimated on the fly later: but once again, the comments on this are not deep precise enough to convince me."
>
> Some motivation on the overall problem is provided in the 2nd section of this response. In regards to the question on the dependence on $\Sigma, \Phi, \delta$ and their motivation, indeed, it is a good question to ask why would we not define an estimator $w$ to be achievable simply if there exists a function $f$ such that $w = f(X,y)$. Why do we first look at the larger class of interpolators, where $w = f(X,y,\Sigma,\Phi,\delta)$, and then try to approximate $\Sigma,\Phi,\delta$?
>
> The answer consists of several reasons. Firstly, the optimal interpolator which can be written as $w = f(X,y,\Sigma,\Phi, \delta)$ provides a fundamental lower limit to performance of an achievable interpolator (as the excess risk depends on $\Sigma, \Phi, \delta$). Thirdly, sometimes a data scientist has access to some prior information about the regression problem which they can incorporate into an estimate of $\Sigma, \Phi, \delta$ (for example, they may know that the components of $X_i$ are independent and hence $\Sigma$ is diagonal) and in such cases, it is relevant to consider a broader class than $w$ such that $w = f(X,y)$.

---

> > ### Comment · Reviewer_hWGm · 2021-08-26
> > **Answer**
> >
> > Thanks you for the detailed rebuttal.
> >
> > I was totally fine with the *motivations* of the paper (and found it very nice), and wanted to now if the authors had *intuitions* behind the technical Lemma they states for:
> > - (i) shape of the estimator
> > - (ii) *preconditioned* gradient descent (sorry for the typo)
> > - (iii) difference with minimum norm
> >
> > It appears that giving intuition behind the (i) and (ii) parts is difficult. But I am quite convinced by (iii) ! The rebuttal being as clear as the paper, added to the fact that I am very convinced that the authors really care about the clearness of their results, I will change my grade to 6, encouraging such clear papers to be accepted !

---

> > > ### Author Response · Authors · 2021-08-26
> > > **Thank you**
> > >
> > > Thank you! We are very glad that you appreciated our effort.

---

### Official Review · Reviewer_472b · 2021-07-16

**Rating:** 6
**Confidence:** 4

**Summary:**

This paper is concerned with inducing a certain type of best possible interpolator in linear regression. It is based on the notion of best possible interpolator, defined in Muthukumar et al., 2019., which is the best estimator (i.e., with minimum risk) among the class of interpolators. Based on that, the authors define a new theoretical estimator, the best estimator among interpolators that are also linear in the response variable.  They manage to induce a closed form of this estimator and prove that it is the limit of preconditional gradient descent under a specific initialization. In addition, using approximations for quantities that are usually unknown in practice (e.g.  population covariance matrix, signal to noise ratio), they turn this theoretical estimator into a fully empirical one and compare its performance to other frequently used estimators, showing that this estimator performs arbitrarily better in certain cases. Finally, they extend this notion of best possible theoretical estimator to the random features setting, where they again induce a closed form for this estimator and prove that it is the limit of preconditioned gradient descent under a specific initialization.


**Limitations And Societal Impact:**

In line:
-64  I don't think that Mourtada deals with interpolators in this paper.
-129 A comment regarding the subscript in the expectation may add some more clarity, as some times this means expected value given this quantity, or expected value given everything else other than this quantity
- 132 "In linear regression (when d>n), minimizing the..."
-480  Maybe you could start with a general W that belongs in this class instead of Wo, as equation (17) seems a bit strange.


**Main Review:**

The paper is interesting to read. It belongs to the long list of papers that are concerned with the overparametrized estimation, but instead of trying to explain why this type of estimation generalizes so well, it aims to find the best linear interpolator in the linear regression setting. By restricting the class of possible estimators in the definition by Muthukumar et al., 2019, they manage to turn a theoretical device into a fully observable estimator. They also define a similar notion of optimal estimator for a more complex model and induce a closed form of this estimator, something that may suggests that we could try to induce this type of optimal estimators for even more complex model like neural networks. There is also an interesting example, where we can see that a frequently used interpolator, the minimum-norm interpolator, has arbitrarily bad generalization. Finally, these examples combined with the results related to preconditional gradient descent highlight nicely the importance of initialization.

The paper could be of greater significance if
a) the construction of the fully-empirical estimator would not be restricted to the isotropic regime
b) (as they note) it included a precise guarantee on the approximation error of the estimation of the covariance matrix Σ
c) instead of performing an empirical analysis for special cases, they could manage to prove a theoretical result that bounds the generalization error of the fully-empirical estimator w.r.t. the generalization error of the theoretical optimal response-linear achievable interpolator.

In addition, I think that some points in the beginning of the proof of Proposition 1 could be written with more clarity.

**Time Spent Reviewing:**

15

---

> ### Author Response · Authors · 2021-08-11
> **Response to reviewer 472b**
>
> We thank the reviewer for spending 15 hours reviewing the paper and provided a detailed review with precise comments and questions.
>
> We agree with the points that the reviewer makes. We provide some further explanations for a), b) and c) below.
>
> ## a) Empirical estimator in non-isotropic regimes
> The reviewer states:
>
> "The paper could be of greater significance if the construction of the fully-empirical estimator would not be restricted to the isotropic regime".
>
> Yes, the experiments regarding the performance of the fully empirical estimator were illustrated only in the isotropic regime in our original paper (Figures 1,2,3 and 4). However, its *construction* is not restricted to the isotropic regime. It will only perform less well in a regime that is not isotropic. Below, we provide new experiments in two non-isotropic regimes. First, let us explain what we mean by saying that the *construction* of the fully empirical estimator is not restricted to the isotropic regime.
>
> Say $\Phi \ne I_d$, so that the prior is not isotropic. If one has no prior knowledge of what $\Phi$ is, then they may still construct the empirical interpolator
>
> $w_{Oe}  =  \bigg(\frac{\delta_e}{d}X^T  +  {{\Sigma_e}^{-\frac{1}{2}}}(X{{\Sigma_e}^{-\frac{1}{2}}})^\dagger\bigg)\bigg( I_n  +  \frac{\delta_e}{d}XX^T\bigg)^{-1}  y, &emsp; &emsp; (1)$
>
> as described in Section 4.1, which uses no knowledge other than the training data $X$ and $y$ (not even the prior $\Phi$). Yes, if we knew $\Phi$, or had some prior information about $\Phi$, then we can incorporate this information into an estimate $\Phi_e$ and use the fully empirical approximation
>
> $w_{Oe\Phi_{e}}  =  \bigg(\frac{\delta_e}{d}\Phi_e X^T  +  {{\Sigma_e}^{-\frac{1}{2}}}(X{{\Sigma_e}^{-\frac{1}{2}}})^\dagger\bigg)\bigg( I_n  +  \frac{\delta_e}{d}X\Phi_eX^T\bigg)^{-1}  y, &emsp; &emsp; (2)$
>
> which is likely to perform better than if we used $\Phi_e = I_d$ as in (1). However, we can *always* construct $w_{Oe}$ (no matter what the prior is) with only the knowledge of the training data $X$ and $y$. It is only a matter of how well it will approximate $w_O$.
>
> The question whether using $w_{Oe}$ with $\Phi_e = I_d$ is suitable when $\Phi \ne I_d$ is equivalent to the question why one would use the (commonly used) minimum-norm interpolator $w_{\ell_2} = X^T(XX^T)^{-1}y$ instead of the interpolator with optimal bias $w = \Phi X^T(X\Phi X^T)^{-1}y$ (for the latter, see Appendix A.3, A.4 of our paper, and Amari et al.).
>
> To further convince the reader, we provide an experiment identical to that of Figure 4 in the paper, with the exception of having a prior where $\Phi$ is in the autoregressive regime. That is
>
> $ \Phi_{ij} = \rho^{|i-j|}$
>
> for all $i,j\in\\{1,\dots, d\\}$, where $\rho = 0.5$. The population covariance matrix $\Sigma$ is in the exponential regime (as described in A.7 of the paper), where $\Sigma_{ii} = -\text{log}(1 - \frac{i}{d+1})$ and the off-diagonal entries are zero.
>
> In Figure 9 (attached in the following link: https://drive.google.com/drive/folders/1lU7cEMjAU5atZJRnmsyhgv6FcoTGn_iz?usp=sharing ) we plot $E_{\xi}r(w)$ (points) for $w \in \\{w_{O}, w_{Oe}, w_{Oe\Phi}, w_{b}\\} $ with $d = \lfloor \gamma n \rfloor, r^2 = 1, \sigma^2 = 1, n = 2000$. Here, $w_{Oe}$ is defined as in (1) and $w_{Oe\Phi}$ as in (2), where the latter uses knowledge of the true covariance $\Phi$. We see that the fully empirical interpolator $w_{Oe}$ performs very similarly to $w_{O}$ and $w_{Oe\Phi}$.
>
> We add another experiment (in Figure 10 of the above link), where here the population covariance is autoregressive, $ \Sigma_{ij} = \rho^{|i-j|}$ with $\rho = 0.5$ and the prior covariance matrix satisfies $\Phi = \Sigma^{-1}$. This corresponds to the 'hard' prior regime considered in Asymptotics of Ridge (less) Regression under General Source Condition by Richards et al.. Here, we plot $E_{\xi}r(w)$ (points) for $w \in \\{w_{O}, w_{Oe}, w_{Oe\Phi}, w_{b}\\} $ with $d = \lfloor \gamma n \rfloor, r^2 = 1, \sigma^2 = 1, n = 2000$. $w_{Oe}$ is defined as in (1) and $w_{Oe\Phi}$ as in (2), where the latter uses knowledge of the true covariance $\Phi$. Again, we see that the fully empirical interpolator $w_{Oe}$ performs very similarly to $w_{O}$ and $w_{Oe\Phi}$.
>
> Comment: We could not include the plot directly in here, but we consulted NeurIPS and we are allowed to upload it as an anonymized google drive link.
>
> ## b) Precise guarantee on the approximation error
> The reviewer states:
>
> "The paper could be of greater significance if (as they note) it included a precise guarantee on the approximation error of the estimation of the covariance matrix $\Sigma$."
>
> Yes, a precise theoretical guarantee on the approximation of $\Sigma$ would be beneficial. First, we are not confident whether this is possible for a general $\Sigma$. Assuming that it is possible, we believe that this would be a paper on its own. We believe our current contribution already sheds light on some important connections to optimality that have not been previously unveiled/investigated, and it can serve to promote further research on extensions and related questions.
>
> ## c) Precise guarantee on the generalization error
> The reviewer states:
>
> "The paper could be of greater significance if instead of performing an empirical analysis for special cases, they could manage to prove a theoretical result that bounds the generalization error of the fully-empirical estimator w.r.t. the generalization error of the theoretical optimal response-linear achievable interpolator."
>
> Yes, a theoretical result that bounds the generalization error of the fully-empirical estimator w.r.t. the generalization error of the optimal response-linear achievable interpolator would be ideal. However, to be able to do this, one would first need to have the theoretical guarantee on the approximation of $\Sigma$, which is a significant problem. Hence this reduces to the problem in b).
>
> ## Minor comment
>
> The reviewer states:
>
> "64 I don't think that Mourtada deals with interpolators in this paper."
>
> In the paper Exact minimax risk for linear least squares, and the lower tail of sample covariance matrices by J.Mourtada, it is stated and proved that
>
> "the ordinary least squares estimator is exactly minimax optimal in the well-specified case for every distribution of covariates",
>
> where the ordinary least squares estimator is the minimum-norm interpolator.

---

> > ### Author Response · Authors · 2021-09-06
> > **Follow up**
> >
> > Dear Reviewer,
> >
> > Just a quick follow up regarding our response to your comments. We think that your point in a) is very good and the answer to it will provide a contribution to the paper's technical quality. Please let us know your opinion on our answer.
> >
> > Moreover, If you have any other outstanding concerns, do let us know as we can provide additional clarification. Otherwise, if you have no other concerns, we would like to kindly ask whether you would provide an update of the review in light of this.
> >
> > Best,
> > Authors

---

### Official Review · Reviewer_JMam · 2021-07-16

**Rating:** 7
**Confidence:** 4

**Summary:**

This paper studies the generalization error of a class of interpolator in linear regression and identifies conditions for optimality. They consider a setting where there is a true linear model generating the data, whose parameters are drawn from a Gaussian distribution with zero-mean and covariance Phi. In this setting they consider a natural class of estimators that are a function of the data, the population covariance matrix, the prior covariance matrix Phi and the signal to noise ratio. Among this class of estimators they calculate a closed-form expression for the optimal interpolator, with respect to the expected excess risk and show that there are certain settings where the expected excess risk of this estimator can be much lower than the risk of the OLS solution. They also provide an extension of their results to the random features model.

**Limitations And Societal Impact:**

The authors do a good job of describing the potential negative impact of their work.

**Main Review:**

The authors study an interesting problem on a topic of much recent interest---benign overfitting of linear models. While the assumptions made on the problem are pretty stringent, the authors provide a clean exposition of their results which I appreciated as a reader.

Here are some questions/comments about the paper:

1. The authors mention the results by Amari et al. (2021), who they mentioned studied estimators whose asymptotic bias or variance is optimal. How do the techniques and setting used in this paper differ from theirs? Can the techniques used there also be used here but instead to balance the bias and the variance optimally? A discussion regarding this would be helpful to the reader.

2. On ln 91 the authors say an interesting question that they study here is if minimum norm interpolant is always close to optimal, and show that this is not the case. But I think this question has already partially been answered given past work of Bartlett et al. (2020). Given their results, it is easy to construct cases, for example, when population covariance matrix is isotropic, where the minimum norm interpolant is not consistent but the ridge regression solution is. So, I would urge the authors to provide some more nuanced context and discussion here.

3. Regarding the assumptions both the noise xi and the true parameter w* are assumed to be drawn from a Gaussian distribution. Can this be relaxed? Further, can the results of this paper (Proposition 1 in particular) be extended to hold in the case where w* is not mean zero?

4. In the experiments described in Section 4.1, where w_O is approximated by w_{O_e}, what is the structure of the population covariance matrix? Is it isotropic or close to it? Or does it have a heavy tailed structure similar to one that was shown to be necessary by Bartlett et al. for the minimum norm solution to be consistent? Further, empirically, was the estimate Sigma_e close to Sigma? A plot to verify this would be nice.


=================== Post rebuttal comments ===================

Firstly, I would like to thank the authors for their very carefully response to the questions raised by all of the reviewers. I continue to support acceptance of this paper post the rebuttal period. In my view, I think it would be nice if the authors incorporate the many helpful suggestions raised in all of the reviews in a subsequent version.


**Time Spent Reviewing:**

2

---

> ### Author Response · Authors · 2021-08-11
> **Response to reviewer JMam**
>
> We thank the reviewer for their feedback and very relevant questions which will help us to improve our manuscript.
>
> ## Comparison with Amari et al.
>
> ### Comparison of setting and technique
>
> The reviewer asks:
>
> "The authors mention the results by Amari et al. (2021), who they mentioned studied estimators whose asymptotic bias or variance is optimal. How do the techniques and setting used in this paper differ from theirs? "
>
> Their setting looks at interpolators achieved as the limit of preconditioned gradient descent in linear regression (preconditioned with some matrix $P$) and initialized at $0$. Such interpolators can be written as $w = PX^T(X PX^T)^{-1}y$. For such interpolators, they compute the risk of $w$, separate the risk into a variance and a bias term and using random matrix theory they find what the variance and bias terms converge to when $d\to\infty, n \to\infty$ in a way such that $d/n\to\gamma\geq 1$. For these calculations to hold it is required to introduce some assumptions. One has to assume that $d/n\to\gamma\geq 1$ as $d\to\infty, n \to\infty$, that the spectral distribution of $(\Sigma_d)_{d\in\mathbb{N}}$ converges weakly to a distribution supported on $[0,\infty)$ and that the eigenvalues of the population covariance matrices are uniformly upper and lower bounded. Then after obtaining the limiting variance and bias, they prove which matrices $P$ minimize these limits *separately* (not their sum - the overall asymptotic risk).
>
> Our results are stronger and more general and our view approaches the problem from the other direction. That is, we do not apriori assume the interpolator is the limit of any algorithm, we directly look at which interpolator minimizes the risk as a whole (not bias and variance seprately), our results hold for every finite $d\geq n$ (we do not assume $d\to\infty, n \to\infty$,$d/n\to\gamma\geq 1$) and we do not put assumptions on the eigenvalues and spectral distribution of the population covariance $\Sigma$. We assume that the interpolator is a linear function of the response variable $y$, as is the case for many estimators in statistics (e.g. ridge regression and all the interpolators in Amari et al.).
>
> In particular, we can recover the results of Amari et al. as a special case of our Proposition 1. If we take the signal-to-noise ratio $\delta\to 0$ (by taking $r^2\to 0$) in Proposition 1, we obtain the matrix $P$ which achieves optimal variance in the paper of Amari et al. and if we take $\delta\to \infty$ (by taking $\sigma^2\to 0$) in Proposition 1, we obtain the matrix $P$ which achieves optimal bias in their paper. These results still hold for finite $n$ and $d$ (and hence also in the limit).
>
> Moreover, Propositon 2 then shows that our interpolator, which minimizes the overall risk (not variance and bias separately), is the limit of preconditioned gradient descent, but with non-trivial initialization (in Amari et al. they considered only $0$ initialization, so they are unable to recover the interpolator with minimal overall risk).
>
> The methodology is also different. Their technique involves calculating limits using random matrix theory while our technique involves minimizing a convex function of rectangular matrices.
>
> ### Can the techniques of Amari et al. be applied in this paper?
>
> The reviewer asks:
>
> "Can the techniques used there also be used here but instead to balance the bias and the variance optimally? A discussion regarding this would be helpful to the reader."
>
> One can see that the techniques used there cannot be used for achieving the optimal balance of bias and variance, because in their analysis they consider only interpolators of the form $w = PX^T(X PX^T)^{-1}y$ for some matrix $P$. Our interpolator is not of this form. However, it is of the form $w = PX^T(X PX^T)^{-1}(y - Xw_0) + w_0$ for a particular initialization $w_0$. One would have to then optimize not only with respect to $P$ (as they do in Amari et al.), but also with respect to $w_0$. It is not clear to us how their technique could be adapted and extended to also cover optimization over $w_0$.
>
> ## Divergence of generalization within class of interpolators.
> The reviewer states:
>
> "On ln 91 the authors say an interesting question that they study here is if minimum norm interpolant is always close to optimal, and show that this is not the case. But I think this question has already partially been answered given past work of Bartlett et al. (2020). Given their results, it is easy to construct cases, for example, when population covariance matrix is isotropic, where the minimum norm interpolant is not consistent but the ridge regression solution is. So, I would urge the authors to provide some more nuanced context and discussion here."
>
> Yes, what the reviewer stated is true. However, our aim is not to illustrate that the minimum-norm interpolant can generalize arbitrarily worse than other, *regularized*, solutions (in fact, optimally-tuned ridge regression provably generalizes at least as good as the minimum-norm interpolator for large $d\geq n$ (the proof holds under specific assumptions)). This is indeed well-known. Our aim is instead to illustrate the phenomenon of arbitrary large difference in generalization *within the class of interpolators* (with the minimum-norm interpolator serving as a benchmark).
>
> Moreover, we show that this phenomenon can be a result of both:
>
>  - the implicit biases of different optimization algorithms (as shown by Figure 1 in connection with Proposition 2),
>  - the choice of initialization of a single optimization algorithm (as shown by Figure 2).
>
> ## Assumptions
> The reviewer asks:
>
> "Regarding the assumptions both the noise xi and the true parameter w* are assumed to be drawn from a Gaussian distribution. Can this be relaxed? Further, can the results of this paper (Proposition 1 in particular) be extended to hold in the case where $w^{\star}$ is not mean zero?"'
>
> Yes, this can be relaxed. The only assumptions we need are that the first and second moments of $\xi_i, w^{\star}$ are finite and that $\xi_1,\dots,\xi_n$ are $\text{i.i.d.}$ with variance $\sigma^2$ and centered. Proposition 1 holds the same even if $w^{\star}$ is not mean $0$. In our original submission, we have followed the same (Gaussian) setup typically considered in similar literature. For the camera-ready version, we will write our results in the most general version. Thank you.
>
> ## Empirical covariance
> The reviewer asks:
>
> "In the experiments described in Section 4.1, where $w_O$ is approximated by $w_{O_e}$, what is the structure of the population covariance matrix? Is it isotropic or close to it? Or does it have a heavy tailed structure similar to one that was shown to be necessary by Bartlett et al. for the minimum norm solution to be consistent? Further, empirically, was the estimate $\Sigma_e$ close to $\Sigma$? A plot to verify this would be nice."
>
> In the examples of the paper (in the main text as well as supplementary material), we used three regimes of the population covariance: the strong weak features model,
>
> $\Sigma = \text{diag}(\rho_1,\dots, \rho_1, \rho_2, \dots, \rho_2)\in \mathbb{R}^{d\times d},$
>
> the autoregressive regime defined by
>
> $
>     \Sigma_{i,j} = \rho^{|i-j|}
> $
>
> for all $i,j \in \\{1,\dots,d\\}$ and some $\rho\in(0,1)$, and finally, an exponential regime
> where the eigenvalues of $\Sigma$ are evenly spaced quantiles of the standard exponential distribution. Namely,
>
> $
>     \Sigma_{i,i} = -\text{log}(1-p_i),
> $
>
> where $p_i = i/(d+1)\in(0,1)$ for $i\in\{1,\dots,d\}$. The off-diagonal entries are $0$.
>
> The eigenvalues of the autoregressive covariance matrix can be shown to be
>
> $\lambda_k = \frac{1-\rho^2}{1-2\rho \text{cos}(\theta_k) + \rho^2},$
>
> where
>
> $\frac{(k-1)\pi}{n+1}<\theta_k <\frac{k\pi}{n+1},$
>
> as stated on page 182 of Asymptotic distribution of the spectra of a class of generalized Kac-Murdock-Szegő matrices by William Trench. The eigenvalues of the other two matrices (strong weak features regime and exponential regime) are straightforward from their definition.
>
> Experiments with the autoregressive and exponential regimes are in the supplementary material. As noted in the supplementary material, these three regimes of covariance matrices have a notable sparsity structure.
>
> Moreover, as the reviewer requested we provide plots to verify how close $\Sigma_e$ was to $\Sigma$ in the experiments. We perform experiments with exactly the same setup as in Figures 1,3,4, but we plot the spectral norm (largest singular value) of $\Sigma_e - \Sigma$. The plots can be seen in Figures 6,7,8 in the following anonymized link: https://drive.google.com/drive/folders/1lU7cEMjAU5atZJRnmsyhgv6FcoTGn_iz?usp=sharing .
>
> We can see that the empirical estimate was close to the population covariance matrix.

---

> > ### Author Response · Authors · 2021-09-06
> > **Follow up**
> >
> > Dear Reviewer,
> >
> > Thank you for your comment regarding our response (in section "Post rebuttal comments" of the original review). We will certainly incorporate the suggestions raised by the reviewers in the very next version of the paper. The comments illustrate important points which are certain to improve the technical quality of the paper.
> >
> > Moreover, If you have any other outstanding concerns which could further improve the assessment of the paper, do let us know as we are happy to provide additional clarification.
> >
> > Best,
> > Authors

---

### Decision · Program_Chairs · 2021-09-27

**Decision:**

Accept (Poster)

**Comment:**

This is a good paper and I recommend to accept it. There are plenty of points in the reviews that suggest ways to improve the paper and I recommend that the authors follow up on these.